# Federated EndoViT: Pretraining Vision Transformers via Federated Learning on Endoscopic Image Collections

**Max Kirchner**[1] (iD)                                          MAX.KIRCHNER@NCT-DRESDEN.DE
**Alexander C. Jenke**[1] (iD)                          ALEXANDER.JENKE@NCT-DRESDEN.DE
**Sebastian Bodenstedt**[1,2] (iD)                SEBASTIAN.BODENSTEDT@NCT-DRESDEN.DE
**Fiona R. Kolbinger**[3,4,5] (iD)                                        FKOLBING@PURDUE.EDU
**Oliver L. Saldanha**[5,6] (iD)                        OLIVER.SALDANHA@DKFZ-HEIDELBERG.DE
**Jakob N. Kather**[5,6,7] (iD)                      JAKOB_NIKOLAS.KATHER@TU-DRESDEN.DE
**Martin Wagner**[2,3] (iD)                                             MARTIN.WAGNER@UKDD.DE
**Stefanie Speidel**[1,2] (iD)                            STEFANIE.SPEIDEL@NCT-DRESDEN.DE

[1] *Translational Surgical Oncology, National Center for Tumor Diseases (NCT), NCT/UCC Dresden, a partnership between DKFZ, Faculty of Medicine and University Hospital Carl Gustav Carus, TUD Dresden University of Technology, and Helmholtz-Zentrum Dresden-Rossendorf (HZDR), Germany.*

[2] *Centre for Tactile Internet with Human-in-the-Loop (CeTI), TU Dresden, Germany*

[3] *Visceral, Thoracic and Vascular Surgery, Faculty of Medicine and University Hospital Carl Gustav Carus, TU Dresden, Germany*

[4] *Weldon School of Biomedical Engineering, Purdue University, West Lafayette, IN, USA*

[5] *Else Kroener Fresenius Center for Digital Health (EKFZ), Faculty of Medicine and University Hospital Carl Gustav Carus, TU Dresden, Germany*

[6] *Medical Oncology, NCT, University Hospital Heidelberg, Germany*

[7] *Medicine I, Faculty of Medicine and University Hospital Carl Gustav Carus, TU Dresden, Germany*

**Editors:** Accepted for publication at MIDL 2026

## Abstract

**Purpose:** Data privacy regulations hinder the creation of generalizable foundation models (FMs) for surgery by preventing multi-institutional data aggregation. This study investigates federated learning (FL) as a privacy-preserving solution to collaboratively train robust surgical FMs. **Methods:** We introduce Federated EndoViT (FL-EndoViT), a federated framework that validates the Masked Autoencoder (MAE) pretraining strategy in a decentralized surgical setting. To ensure convergence under severe data heterogeneity, the architecture integrates adaptive Sharpness-Aware Minimization (FedSAM). Pretrained on the large-scale Endo700k dataset, FL-EndoViT is evaluated against a centralized baseline on different tasks including scene segmentation, action recognition, and phase recognition. **Results:** FedSAM is critical for successful pretraining, overcoming the convergence failures of standard federated methods. The resulting FL-EndoViT performs comparably to its centralized counterpart, with significant advantages in data-scarce, high-resolution segmentation and generalization to new surgical events. We also establish that full, end-to-end fine-tuning is necessary for optimal performance. **Conclusion:** This work validates FL with adaptive optimization as a viable paradigm for creating robust, privacy-preserving surgical FMs. Our findings provide a scalable framework for collaborative Surgical Data Science and underscore the optimizer's critical role in handling data heterogeneity. Future work should explore video-based models to incorporate spatiotemporal dynamics.

**Keywords:** Endoscopic Video Analysis, Federated Learning, Foundation Models, Surgical Data Science, Vision Transformers

## 1. Introduction

The transformative potential of modern surgery relies heavily on the ability to convert complex intraoperative data into actionable clinical intelligence. This pursuit defines Surgical Data Science (SDS), a field where the deployment of scalable, collaborative frameworks is contingent upon achieving both model robustness and rigorous privacy preservation (Maier-Hein et al., 2022). Within this landscape, Foundation Models (FMs) have emerged as a powerful paradigm for addressing diverse downstream tasks by learning generalized representations from extensive datasets (Bommasani et al., 2021). Yet, a critical paradox remains; despite their theoretical versatility, FMs developed on homogeneous or limited data frequently fail to generalize across distinct patient populations and institutions, thereby hindering their reliable application in clinical settings (Jogan et al., 2024). Homogeneous data lacks the variability that stems from differences in patient anatomy, surgical techniques, equipment, and clinical protocols in different institutions, and thus hinder the generalization capabilities of FMs (Jogan et al., 2024). To compound these challenges, data protection regulations such as the Health Insurance Portability and Accountability Act (HIPAA) (Rights , OCR) and the General Data Protection Regulation (GDPR) (noa, 2016) impose strict constraints on patient data sharing, making cross-institutional data aggregation legally and organizationally complex. Federated Learning (FL) offers a solution for this predicament by enabling collaborative model training across medical institutions without data sharing, maintaining privacy while leveraging diverse datasets (McMahan et al., 2017; Bommasani et al., 2021).

Recently, hybrid frameworks integrating FL and FMs have been introduced for medical imaging, combining the robustness of FMs with the privacy-preserving properties of FL. For instance, FedFMS (Liu et al., 2024) adapts the Segment Anything Model (Kirillov et al., 2023) for federated segmentation across diverse modalities (MRI, nuclei, fundus photography, X-ray). Similarly, FedEFM (Do et al., 2025) employs knowledge distillation with differentiable Earth Mover's Distance (EMD) to address label or data mismatches in federated X-ray settings, while FedKim (Wang et al., 2024) utilizes adaptive knowledge injection for MRI and CT foundation models. Despite these advances in radiology, research within the surgical domain remains limited. Notable exceptions include FedCy (Kassem et al., 2023), which was developed specifically for Surgical Phase Recognition (SPR) but lacks the breadth of a general-purpose foundation model.

The absence of robust, privacy-preserving architectures that integrate Federated Learning (FL) and FMs within the surgical domain stems primarily from two critical data impediments: the scarcity of large-scale, multi-centric surgical datasets and the intrinsic heterogeneity of surgical data (Maier-Hein et al., 2022). Compounding this complexity, surgical video data manifests substantial variability regarding operative technique, recording parameters, clinical environments, and temporal dynamics (Eckhoff et al., 2024; Kolbinger et al., 2025). These challenges precipitate a fundamental inquiry: Can FM trained via FL achieve performance parity with those trained in traditional centralized settings, and what optimization strategies are required to ensure robust convergence?

To address this hypothesis and bridge the existing gap, this study introduces the Federated EndoViT (FL-EndoViT) framework. This architecture utilizes the Endo700k dataset collection (Batić et al., 2024) to pretrain a FM within a decentralized paradigm. The pro-

posed framework synergizes the EndoViT vision transformer (Batić et al., 2024) with a federated Masked Autoencoder (MAE) pretraining strategy. Furthermore, the incorporation of adaptive Sharpness-Aware Minimization (SAM) (Caldarola et al., 2022) and Stochastic Weight Averaging (SWA) (Izmailov et al., 2018) enables the model to effectively mitigate the adverse effects of non-Identically and Independently Distributed (non-IID) surgical data. Crucially, we identify that the efficacy of federated training for surgical FMs is contingent upon the optimization strategy employed. Our experiments reveal that standard federated aggregation fails to converge due to the extreme data heterogeneity characteristic of multi-institutional surgical environments. Therefore, we demonstrate that adaptive FedSAM is critical for neutralizing these shifts and facilitating successful MAE pretraining.

Following pretraining, the architecture undergoes fine-tuning across a spectrum of downstream tasks, including Surgical Scene Segmentation (SSS), Action Triplet Recognition (ATR), Surgical Phase Recognition (SPR), and classification tasks for action, blood, and smoke detection. FL-EndoViT delivers competitive general performance while excelling in specialized contexts; specifically, it achieves parity with centralized models in standard recognition tasks like SPR and ATR, yet demonstrates superior efficacy in clinically distinct scenarios. Notably, the federated model exhibits a distinct advantage in data-scarce, high-resolution SSS. Furthermore, regarding generalization to unseen surgical domains, FL-EndoViT displays enhanced proficiency in identifying rare events, such as bleeding and smoke, indicating the acquisition of more diverse and robust feature representations.

Nevertheless, the analysis highlights specific performance trade-offs. The superiority of the federated model is task-dependent; the centralized model holds an advantage in low-resolution settings and the classification of majority-class actions. Additionally, achieving optimal performance necessitates comprehensive, computationally intensive end-to-end fine-tuning. Crucially, however, this study benchmarks against an idealized centralized model trained on fully pooled datasets. This scenario is rarely achievable in clinical practice due to strict regulatory barriers. In a real-world setting, where a centralized model would be restricted to smaller, isolated datasets, we expect its performance to degrade significantly, thereby widening the performance gap in favor of the federated approach.

In summary, our main contributions are:

1. **Validation of Federated Surgical Foundation Models:** We provide a systematic validation of FL-EndoViT on the decentralized Endo700k dataset, proving that privacy-preserving MAE pretraining is feasible and effective for surgical videos.

2. **Optimization for Surgical Heterogeneity:** We empirically demonstrate that standard federated aggregation (e.g., FedAvg) fails to converge on heterogeneous surgical video data. Hence, we establish that adaptive Sharpness-Aware Minimization (FedSAM) is a necessary component to overcome these non-IID challenges and stabilize FM training.

3. **Superior Generalization:** Beyond achieving parity with centralized baselines, we show that FL-EndoViT outperforms them in critical scenarios, specifically improving segmentation in data-scarce regimes and generalizing better to rare surgical events.

4. **Reproducibility:** To facilitate future research, we make our training code publicly available at https://github.com/KirchnerMax/FL-EndoViT.

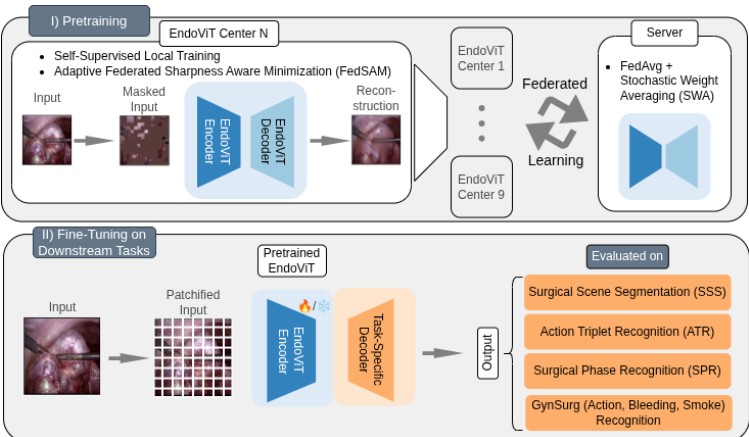

Figure 1: **Federated EndoViT Framework.** (I) Federated self-supervised MAE pre-training (75% masking) utilizing client-side Adaptive FedSAM and server-side SWA for non-IID generalization. (II) Fine-tuning the pretrained encoder backbone on diverse surgical downstream tasks.

Collectively, through its generalizable and privacy-preserving design, the proposed framework establishes a scalable paradigm for collaborative SDS, enabling multi-institutional data utilization while adhering to strict data protection standards. Consequently, this innovation advances predictive analytics, workflow optimization, and outcome assessment across diverse clinical contexts, ultimately elevating the standard of surgical care and patient outcomes.

## 2. Methodology

This work introduces FL-EndoViT, a federated FM framework designed to learn robust surgical representations from decentralized, heterogeneous data sources. As illustrated in Figure 1, the approach consists of two distinct phases: federated self-supervised pretraining and centralized downstream fine-tuning.

### 2.1. Federated Pretraining with Adaptive Optimization

In this work, a FL environment is simulated using the Endo700k dataset collection (Batić et al., 2024), partitioned across nine distinct clients. Clients possess unique sub-datasets across varying domains (e.g., colorectal, urologic) and modalities (laparoscopic vs. robotic), creating a highly non-IID distribution mimicking real-world clinical silos. Within this decentralized framework, the EndoViT Vision Transformer (Batić et al., 2024) functions as the backbone, utilizing a MAE objective where the model learns to reconstruct the original image from inputs where 75% of the original image is masked out. To mitigate convergence issues common in such heterogeneous settings, where client models often converge toward sharp, non-generalizable local minima, the Adaptive Federated Sharpness-Aware Minimization (FedSAM) optimization strategy (Caldarola et al., 2022) is employed to avoid

this behavior. Specifically, this optimizer is applied during local client updates to minimize both loss value and sharpness, thereby guiding the optimization toward flatter, more robust minima. This is complemented by a client-side stabilization strategy. SWA (Caldarola et al., 2022; Izmailov et al., 2018) is implemented on the server during the final pretraining epochs to aggregate global model weights, further smoothing the loss landscape and enhancing generalization.

## 2.2. Downstream Task Evaluation

Following pretraining, the encoder is fine-tuned on four downstream tasks to evaluate feature robustness: SSS on CholecSeg8k (Hong et al., 2020), ATR on CholecT45 (Nwoye et al., 2022; Nwoye and Padoy, 2022), SPR on Cholec80 (Twinanda et al., 2016), and Video Classification on the GynSurg dataset (Nasirihaghighi et al., 2025). The fine-tuning strategy is to fine-tune end-to-end because preliminary ablation studies confirmed that full fine-tuning significantly outperforms frozen-backbone approaches (see Appendix B). Each of the downstream task uses the FL-EndoViT as the backbone and is compared to the baseline model CEN-EndoViT. For each task a different head is attached (see Appendix A.1).

## 2.3. Implementation and Data Splits

The setup was implemented using Flower framework (Beutel et al., 2020). To prevent data leakage between pretraining and downstream tasks, three distinct partitions of the Endo700k dataset collection were established. Any video appearing in a downstream validation or test set was strictly excluded from the pretraining corpus. For the GynSurg evaluation, the ATR-variant of the pretraining set was utilized due to domain similarity. Machine learning configurations are provided in Appendix E.

## 3. Experiments, Results and Discussion

This study begins with an ablation analysis to ascertain the criticality of integrating Fed-SAM into MAE pretraining, aiming to achieve performance metrics equivalent with centralized training. Subsequently, we compared FL-EndoViT and CEN-EndoViT across four distinct surgical tasks, necessitating fine-tuning for optimal model performance. Finally, following the description of the EndoViT fine-tuning methodology, a comparative experiment encompassing both full-shot and few-shot training paradigms was conducted. To establish statistical significance between the models, a paired Wilcoxon signed-rank test was employed. For SSS, the test was conducted at the image level, whereas for other tasks, it was performed at the video level. The null hypothesis, positing comparable model performance, was rigorously tested at a significance level ($\alpha$) of 0.01, thereby demanding more robust evidence for rejection and underscoring the profound significance of the findings.

### 3.1. Federated Pretraining Ablation Study and Results

The efficacy of adaptive FedSAM in bridging the performance gap between federated and centralized pretraining paradigms was examined through a targeted ablation study. Within

this analysis, the integrity of the learned visual representations was inferred from reconstruction loss, quantified by the prevalence of image patches below various Mean Squared Error (MSE) constraints (0.3, 0.1, 0.05, 0.01).

As evidenced in Table 1, standard Federated Averaging (FedAvg) and alternative aggregation schemes exhibit significant performance deficits. To ensure a rigorous and fair comparison, we maintained consistent base hyperparameters across the experiments, specifically keeping the total number of rounds (T), client fraction (C), local epochs (E), local batch size (B), and local learning rate ($\eta$) identical. For the comparative aggregation algorithms, we utilized the standard configurations provided by the Flower framework to represent commonly accepted baselines while specifically tuning algorithm-specific parameters. This included optimizing the fairness parameter ($Q$) for QFedAvg, the trim percentage for FedMedian, server-side momentum for FedAvgM, and the decay constants for FedAdam to ensure each method operated under optimal conditions.

Conversely, the integration of adaptive FedSAM into the framework precipitates a marked improvement, yielding results that approximate the centralized baseline. This performance contrast underscores the critical influence of optimization strategy on federated representation learning. The severe non-IID nature of the Endo700k dataset collection amplifies convergence challenges, particularly for dense, self-supervised MAE tasks where reconstructing pixel values magnifies sensitivity to domain shifts. By explicitly favoring flat minima within the loss landscape, adaptive FedSAM mitigates the instability inherent in averaging models that have converged to differing sharp minima. Additionally, SWA is retained to ensure stability in these heterogeneous settings, aligning with best practices despite negligible immediate impact on validation metrics. Consequently, the FL-EndoViT architecture incorporating both FedSAM and SWA and is designated for subsequent evaluation. While the optimized federated performance closely tracks the centralized benchmark, the residual gap suggests that downstream fine-tuning serves a dual purpose: adapting features to specific tasks and compensating for initial pretraining differentials. Future investigations should therefore prioritize intrinsic evaluations of pretrained features to determine if the model functions as a truly foundational representation or an optimized initialization.

### 3.2. Downstream Tasks

We evaluated the FL-EndoViT backbone by fully fine-tuning the model including the backbone and task-specific head on various downstream tasks. In particular, this analysis covers the SSS, ATR, and SPR tasks in both full and few-shot learning scenarios. Additionally, to further assess generalization to a different surgical domain, we also report performance on the GynSurg classification tasks. To ensure robustness, results for SSS, ATR, and SPR are averaged over three independent technical replications, while the GynSurg tasks were evaluated using 4-fold cross-validation. All performance metrics are reported as mean $\pm$ standard deviation.

SURGICAL SCENE SEGMENTATION

The FL-EndoViT backbone outperforms the centralized baseline in the few-shot high-resolution setting (see Table 2). This advantage is particularly pronounced in data-scarce scenarios; for instance, training on a single video yields an average mIoU of 42.03% for

Table 1: **Pretraining Performance.** Distribution of reconstructed patches (out of 256) below MSE thresholds on Endo700k. SWA results are shown in parentheses. Adaptive FedSAM is the only federated method to approximate the centralized baseline, effectively mitigating non-IID data heterogeneity.

| Method | THRE[1] 0.3 | THRE[1] 0.1 | THRE[1] 0.05 | THRE[1] 0.01 |
|---|---|---|---|---|
| **Centralized Baseline** | 171 (171) | 126 (120) | 92 (90) | 45 (45) |
| **Adaptive FedSAM** (ours) | **168 (166)** | **112 (107)** | **82 (80)** | **39 (38)** |
| **FedAvg** | 67 (67) | 25 (25) | 11 (11) | 0 (0) |
| **FedMedian** | 39 (39) | 5 (5) | 0 (0) | 0 (0) |
| **QFedAvg** ($Q = 2$) | 53 (49) | 8 (7) | 1 (1) | 0 (0) |
| **QFedAvg** ($Q = 0.5$) | 65 (63) | 20 (16) | 6 (4) | 0 (0) |
| **FedAvgM** | 67 (66) | 24 (24) | 11 (11) | 0 (0) |
| **KRUM** | 65 (65) | 23 (23) | 10 (10) | 0 (0) |
| **FedAdam** | 67 (66) | 24 (24) | 11 (11) | 0 (0) |

[1] Threshold below

the federated model compared to 25.68% for the centralized version. This is a significant improvement of 16.35%. This superior performance shows that federated pretraining paradigm, where learning from diverse, decentralized data can compel the development of a versatile feature foundation. This foundation acts as a regularizer, preventing the model from overfitting to limited fine-tuning examples and enabling effective generalization.

Conversely, this trend is reversed in low-resolution scenarios, where CEN-EndoViT generally excels. It is hypothesized that the federated averaging process, while promoting generalization, may inadvertently dampen high-frequency feature representations crucial for interpreting fine details in ambiguous, low-resolution inputs. The centralized model, trained on co-located data, appears to better retain this feature sharpness. However, in both settings, the performance gap diminishes as the volume of fine-tuning data increases (narrowing to a 1.83% difference on the full dataset), indicating that a strong downstream signal allows the model to adapt and overcome initial foundational deficits.

## ACTION TRIPLET RECOGNITION

For ATR, FL-EndoViT outperforms centralized learning in the full fine-tuning setting, achieving a mean Average Precision (mAP) of 40.79% compared to 31.33% (Table 3). This substantial improvement confirms that the federated configuration effectively leverages diverse multi-source data to improve generalization. Conversely, in few-shot scenarios, the performance gap becomes negligible, indicating that both configurations perform similarly under strict data scarcity.

## SURGICAL PHASE RECOGNITION

The TeCNO model follows a two-stage structure: Stage S1 focuses on static feature extraction, while Stage S2 introduces temporal modeling.

Table 2: Few-shot SSS performance (mean IoU-Score) comparison by training set size and resolution. The superior-performing model is indicated in bold, and an asterisk (*) marks statistically significant differences (Wilcoxon paired test, $\alpha = 0.01$).

| Fine-Tuned on | Res. | CEN-EndoViT | FL-EndoViT |
|---|---|---|---|
| 1 Video | Low | **42.87% ± 6.84%*** | 40.50% ± 7.60% |
| | High | 25.68% ± 8.32% | **42.03% ± 6.50%*** |
| 2 Videos | Low | **56.60% ± 5.54%*** | 53.32% ± 5.40% |
| | High | 37.12% ± 3.82% | **56.39% ± 6.39%*** |
| 4 Videos | Low | **61.06% ± 5.62%*** | 58.98% ± 6.25% |
| | High | 45.51% ± 5.61% | **60.34% ± 6.58%*** |
| All Videos | Low | **67.54% ± 8.06%*** | 67.42% ± 8.18% |
| | High | **67.81% ± 8.03%*** | 65.98% ± 8.10% |

Table 3: Few-shot ATR performance (mean average precision) comparison by training set size. The superior-performing model is indicated in bold, and an asterisk (*) marks statistically significant differences (Wilcoxon paired test, $\alpha = 0.01$).

| Fine-Tuned on | CEN-EndoViT | FL-EndoViT |
|---|---|---|
| 2 Videos | **22.61% ± 3.04%** | 21.76% ± 4.30% |
| 4 Videos | 26.00% ± 4.18% | **27.24% ± 4.75%** |
| 8 Videos | **33.16% ± 6.41%** | 32.53% ± 5.71% |
| All Videos | 31.33% ± 4.61% | **40.79% ± 6.65%*** |

In Stage S1, FL-EndoViT and CEN-EndoViT demonstrate comparable efficacy, with only minor differences in F1 scores (Table 4). Performance is largely data-dependent: CEN-EndoViT tends to outperform on limited data, whereas FL-EndoViT matches or exceeds it as data availability increases. These results indicate that both backbones are effective feature extractors, though few-shot performance remains sensitive to the specific choice of fine-tuning datasets.

In Stage S2, the inclusion of temporal components improves performance across all models compared to S1. Crucially, there is no statistically significant difference between FL-EndoViT and CEN-EndoViT. This confirms that the federated backbone captures temporal dependencies as effectively as the centralized model, even when features originate from distributed, multi-institutional data.

Overall, these findings validate that federated backbones match centralized models in both feature extraction and temporal modeling, underscoring the suitability of federated learning for privacy-critical SDS scenarios.

### 3.2.1. GynSurg Video Classification

Table 5 summarizes the GynSurg experiments, which assess how well the models generalize to surgical domains distinct from cholecystectomy. FL-EndoViT achieved the highest mean performance in bleeding recognition (82.48% Acc, 81.60% F1) and smoke recogni-

Table 4: Few-shot SPR performance (Accuracy and F1-Score) comparison by training set size and TeCNO stage. The superior-performing model is indicated in bold, and an asterisk (*) marks statistically significant differences (Wilcoxon paired test, $\alpha = 0.01$).

| Metric | | Accuracy | | F1-Score | |
|---|---|---|---|---|---|
| FT[1] | Stg[2] | CEN-EndoViT | FL-EndoViT | CEN-EndoViT | FL-EndoViT |
| 2 | S1 | **63.83% ± 8.94%*** | 61.30% ± 8.51% | **48.95% ± 8.26%*** | 45.70% ± 6.96% |
| | S2 | **78.13% ± 8.94%** | 76.35% ± 10.53% | 75.15% ± 6.86% | **75.22% ± 8.13%** |
| 4 | S1 | **66.79% ± 9.92%** | 66.23% ± 9.15% | 52.50% ± 10.00% | **53.22% ± 8.94%** |
| | S2 | 79.75% ± 9.55% | **80.07 ± 8.33%** | 77.67% ± 7.50% | **77.89% ± 8.04%** |
| 8 | S1 | **74.04% ± 8.65%*** | 71.96% ± 10.32% | 58.76% ± 9.90% | **60.62% ± 10.02%*** |
| | S2 | 84.34% ± 7.94% | **84.61 ± 8.33%** | 80.73% ± 6.86% | **81.55% ± 6.46%** |
| All | S1 | **81.58% ± 8.02%** | 81.46% ± 7.59% | **71.47% ± 8.46%** | 71.19% ± 8.26% |
| | S2 | 88.37% ± 7.16% | **89.04 ± 7.03%** | 84.48% ± 6.15% | **85.05% ± 5.92%** |

[1] Fine-Tuned on Number of Videos
[2] TeCNO Stage

Table 5: Fine-tuning performance comparison on GynSurg classification tasks (Accuracy and F1-Score). The superior-performing model is indicated in bold, and an asterisk (*) marks statistically significant differences (Wilcoxon paired test, $\alpha = 0.01$).

| FT for[1] | CEN-EndoViT | | FL-EndoViT | |
|---|---|---|---|---|
| | **Accuracy** | **F1-Score** | **Accuracy** | **F1-Score** |
| **Action** | 69.09% ± 7.38%* | 64.43% ± 7.33% | **69.14% ± 5.64%** | **64.98% ± 9.46%*** |
| **Bleeding** | 78.12% ± 11.16% | 73.99% ± 16.65% | **82.48% ± 8.53%*** | **81.60% ± 8.12%*** |
| **Smoke** | 86.44% ± 6.90% | 85.96% ± 6.17% | **86.83% ± 6.96%*** | **86.42% ± 6.39%*** |

[1] Fine-Tuned for

tion (86.83% Acc, 86.42% F1). The action recognition task showed a different pattern. Although FL-EndoViT reached slightly higher mean accuracy and F1 values, a Wilcoxon signed-rank test indicated that CEN-EndoViT achieved statistically stronger and more consistent accuracy on this task ($p < 0.01$). Class-wise analysis (see Appendix 11) further shows that CEN-EndoViT performs more effectively on the majority classes coagulation and rest, whereas FL-EndoViT performs better on minority classes such as needle passing, suction and transection.

These results indicate that both models adapt well but exhibit distinct strengths. The superior and statistically significant performance of FL-EndoViT on bleeding and smoke recognition suggests an advantage in detecting varied or less frequent events. This effect appears to arise not from data characteristics but from the federated training paradigm,

which acts as a strong regularizer. Optimizing a single model across heterogeneous client datasets encourages the learning of robust and domain-invariant representations.

Performance on the action recognition task depends on the evaluation metric. CEN-EndoViT achieves higher accuracy by excelling on high-frequency majority classes, while FL-EndoViT achieves a higher F1-score through more balanced performance and improved recognition of minority classes. This pattern supports the interpretation that the federated model acquires more versatile representations, whereas the centralized model tends to specialize in the most common patterns in the data.

Overall, these findings provide evidence that the federated approach yields a model with strong generalization to new surgical tasks. The performance is highly competitive and, in certain tasks, exceeds that of the centrally trained model. This highlights the robustness of the method and its suitability for practical deployment in settings characterized by substantial data diversity.

### 3.3. Limitations and Future Directions

The FL-EndoViT approach is on par with centralized learning and is performing some times even better. However, the EndoViT approach remains sensitive to data quality and resolution during the adaptation phase. In the SSS task, the FL backbone outperformed the centralized baseline on high-quality data but underperformed on low-quality inputs. We attribute this to the averaging nature of FL. During pretraining, the aggregation of updates creates a strong general representation but tends to smooth out fine-grained details, such as sharp object boundaries, due to the lack of direct access to raw data. When the downstream dataset offers high resolution, it provides the necessary signal to sharpen these boundaries, unlocking the FL backbone's generalization capabilities. Conversely, low-resolution data lacks the fidelity required to refine these smoothed features. Consequently, the federated model struggles to recover precise edge delineations in this setting, causing it to lag slightly behind the centralized baseline, which retains stronger low-level spatial priors from raw-data pretraining.

Another limitation is that the EndoViT two stage approach is resource-intensive because optimal performance necessitates fully fine-tuning both the backbone and the head rather than relying on more efficient frozen-backbone configurations.

Notably is that performance gains also vary by task and accessible training data amount. For example, the advantage of the federated model narrows in few-shot scenarios and is merely comparable in tasks such as SPR. This implies a potential trade-off between standardization and specialization. The improved fairness and homogenization inherent in FedAvg may inadvertently reduce the sensitivity of the model to rare, site-specific events.

Furthermore, the scope of this analysis is concentrated on visceral and abdominal Minimally Invasive Surgery (MIS). While the current architecture establishes a strong baseline within this domain, scaling to distinct surgical fields, such as orthopedic, cardiothoracic, or ocular procedures, presents a valuable avenue for future research. Nevertheless, extending the training data to substantially different fields introduces a higher risk of negative transfer due to significant differences in anatomy, instrumentation, and visual context. Our study purposefully excluded these domains to maintain a controlled setting that isolates the benefits of federated pretraining without introducing uncontrolled domain shifts. To safely

expand to such heterogeneous environments, future translational iterations should explore personalized federated learning mechanisms to selectively weight institutional contributions based on their alignment with the global task. Similarly, integrating dynamic aggregation strategies offers a promising direction for further mitigating heterogeneity. Recent works, such as Dynamic Barlow Continuity (DynBC) (Babendererde et al., 2025), demonstrate that guiding aggregation with adaptive constraints can outperform centralized training in FL settings. Incorporating such dynamic mechanisms into the FL-EndoViT pipeline could effectively safeguard performance as the system scales to increasingly diverse surgical environments.

Finally, future work should explore FL for video-based models in greater depth. As surgical procedures are inherently dynamic, extending FL to architectures that jointly model spatial and temporal features could further advance SDS.

## 3.4. Practical Implications

Constructing a centralized model like CEN-EndoViT with nine datasets represents an idealized benchmark rarely achievable in practice. FL provides a vital alternative for training FMs without pooling sensitive patient data. FL mitigates these barriers and broadens access to diverse data while distributing computational demands.

However, these advantages introduce new challenges. The framework incurs substantial communication overhead and statistical heterogeneity across institutions complicates optimization. To mitigate the risk of negative transfer arising from heterogeneity, our method aims to leverage the adaptive FedSAM optimizer and SWA. Standard optimizers frequently converge to sharp minima that may generalize poorly when client data distributions shift. In contrast, adaptive FedSAM is designed to minimize both the loss value and the loss sharpness by optimizing within a local neighborhood of parameters. Complementing this, SWA averages multiple checkpoints along the optimization trajectory to help smooth the decision boundary. By targeting these flatter regions of the loss landscape, these mechanisms are intended to promote a global model that remains resilient to the erratic gradients induced by non-IID data. For example, our model (116.66M parameters) required ca. 893 MB of bidirectional updates per client per round. Over 15 epochs, this reached an aggregate transfer of ca. 121 GB of model weights, metrics, and metadata. While FL introduces these communication costs, it is a necessary trade-off for the critical advantage of keeping all sensitive patient data local.

Furthermore, real-world deployment requires robust safeguards. One major requirement is capable edge infrastructure. With a computational complexity of 166.6 TMACs (Multiply-Accumulate operations) per image in the forward pass, the model is resource-intensive due to the self-attention mechanisms required for high-resolution surgical frames. An advantage of the federated setup is this computational load is distributed across the centers. However, the load is heavily skewed due to dataset imbalance in our case, with the largest site (HeiCo) performing 47.7% of the operations (see Table 8). Consequently, for practical deployment, we recommend clients utilize at least 32 GB VRAM GPUs and 16-core CPUs, supported by 64 GB system RAM and a 1 Gbps network connection. Equally critical is the system's resilience to intermittent connectivity and asynchronous participation inherent in clinical networks. While our simulation assumed a stable environment to

establish a baseline, our implementation leverages the Flower framework to handle client non-responsiveness. By monitoring minimum fit and evaluation rates, the server effectively manages hardware crashes or network dropouts, ensuring global model aggregation continues provided a threshold of successful updates is met.

Finally, production environments must address the integrity of the model updates themselves. Robustness against adversarial attacks or corrupted data is a primary concern in cross-institutional settings. While this study established a baseline using trusted nodes, real-world deployments face risks ranging from transmission noise to active model poisoning. To mitigate this, our framework supports robust aggregation strategies such as Krum, which filters contributions deviating significantly from the global distribution. Future translational iterations must further harden the system by combining anomaly detection (e.g., Krum and FedSAM) with secure aggregation protocols (e.g., Homomorphic Encryption or Secure Multi-Party Computation (SMPC)) to ensure the integrity of the FL-EndoViT global model in untrusted environments.

Despite these hurdles, FL facilitates multi-institutional learning from diverse anatomies and surgeon styles to improve generalization. Its combination with SSL is particularly promising for reducing annotation costs by exploiting large volumes of unlabeled surgical video.

## 4. Conclusion

This study provides a rigorous validation of FL in training models that generalize across diverse surgical datasets while strictly maintaining privacy. Our results establish that standard optimization methods are insufficient for heterogeneous surgical video data, however, when enhanced with adaptive FedSAM, federated models overcome these convergence failures and achieve strong robustness. Crucially, FL-EndoViT fine-tuned for downstream tasks achieved performance comparable to and occasionally exceeding a centralized model trained on pooled data. This finding validates FL as a viable alternative to centralized training. It offers a path to robust, privacy-preserving models for SDS applications without requiring the impractical aggregation of sensitive institutional data.

## Acknowledgments

This work was co-funded by the European Union through NEARDATA under grant agreement ID 101092644, the German Research Foundation (DFG, Deutsche Forschungsgemeinschaft) as part of Germany's Excellence Strategy – EXC 2050/1 – Project ID 390696704 – Cluster of Excellence "Centre for Tactile Internet with Human-in-the-Loop" (CeTI) of Technische Universität Dresden, and the Federal Ministry of Education and Research of Germany in the programme of "Souverän. Digital. Vernetzt.", a joint project 6G-life with the project identification number 16KISK001K, and by the BMFTR (Federal Ministry of Research, Technology and Space) in DAAD project 57616814 (SECAI, School of Embedded Composite AI) as part of the program Konrad Zuse Schools of Excellence in Artificial Intelligence. The authors acknowledge the financial support by the Federal Ministry of Research, Technology and Space of Germany in the programme of "DigiLeistDAT". Joint project SurgicalAIHubGermany, project identification number: 02K23A112.

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

## Appendix A. Related Work

We leverage the EndoViT model (Batić et al., 2024) as our architectural foundation within a Federated Learning framework, further enhancing optimization via the SAM approach (Caldarola et al., 2022)

### A.1. EndoViT

We selected EndoViT, a Masked Autoencoder (MAE), over contrastive self-supervised learning methods (e.g., MoCo (He et al., 2020), DINO (Ramesh et al., 2023)) or multi-modal

approaches like SurgVLP (Yuan et al., 2025). MAE is particularly advantageous for federated settings because it does not require large negative sample queues, thereby reducing the communication and computational burden. Furthermore, the MAE objective efficiently captures fine-grained spatio-temporal patterns in surgical data, producing representations that are well-suited for end-to-end fine-tuning across heterogeneous institutions.

The EndoViT framework employs a standard MAE strategy to acquire generalizable visual representations. The process involves three key steps (see Figure 1):

1. **Masking:** Each input image is divided into 256 non-overlapping patches, and 75% are randomly masked.

2. **Encoding:** A Vision Transformer (ViT) encoder processes only the visible patches to generate latent representations.

3. **Reconstruction:** A lightweight decoder attempts to reconstruct the masked patches from these representations, minimizing the mean squared error (MSE) between the reconstruction and the original.

This pretraining is conducted on the Endo700k dataset collection (Batić et al., 2024). This large-scale aggregation unifies nine publicly available surgical video datasets to ensure high data heterogeneity across procedures, modalities, and annotation types. The collection includes HeiCo, Cholec80, PSI-AVA, ESAD, LapGyn4, hSDB-Instrument, DSAD, GLENDA, and Surgical Actions 160.

Following pretraining, the backbone is fine-tuned and evaluated on three standard surgical tasks. Previous work indicates that EndoViT matches or surpasses ImageNet-pretrained ViTs on these benchmarks, particularly in data-scarce settings (Batić et al., 2024).

- **Surgical Scene Segmentation (SSS).** We utilize the CholecSeg8k dataset (Hong et al., 2020). The architecture consists of the ViT backbone paired with a Dense Prediction Transformer (DPT) convolutional decoder. Performance is measured by Intersection over Union (IoU).

- **Action Triplet Recognition (ATR).** We utilize the CholecT45 dataset (Nwoye and Padoy, 2022). The encoder is paired with a linear classification head. Following standard protocol, training and evaluation occur on the fifth fold, with mean Average Precision (mAP) as the primary metric.

- **Surgical Phase Recognition (SPR).** We utilize the Cholec80 dataset (Twinanda et al., 2016). The backbone is integrated into the TeCNO model (a Multi-Stage Temporal Convolutional Network) (Czempiel et al., 2020). Evaluation is based on frame-level accuracy.

## A.2. Adaptive Federated Sharpness-Aware Minimization

To address the performance degradation caused by data heterogeneity in Federated Learning, Caldarola et al. (Caldarola et al., 2022) identify the sharpness of the loss landscape as a critical factor: sharper minima correlate with poorer generalization. To mitigate this,

they propose a two-tiered optimization strategy comprising Sharpness-Aware Minimization (SAM) and Stochastic Weight Averaging (SWA).

Sharpness-Aware Minimization (SAM) SAM modifies local training to target "flat" regions of the loss landscape—areas where small perturbations in model parameters produce only negligible changes in loss on the client side. By minimizing both the loss value and its sharpness simultaneously, SAM guides the model toward neighborhoods of uniformly low loss, making the local models less sensitive to specific data variations.

Complementing the client-side optimization, the server employs SWA to aggregate client models. By averaging weights over multiple training epochs, SWA promotes convergence to broader minima. As demonstrated by Izmailov et al. (Izmailov et al., 2018), this trajectory averaging helps mitigate sharp minima, resulting in a global model with superior generalization to unseen data.

Implementation and Results Empirical evaluations demonstrate that integrating these approaches significantly enhances performance across various vision tasks, including semantic segmentation and domain generalization. The specific integration of SAM and SWA into the federated workflow is detailed in Algorithm 1 (adapted directly from (Caldarola et al., 2022)).

## Appendix B. Ablation Study: Full Fine-tuning vs. Frozen Backbone

Table 6: **Impact of Frozen vs. Full Fine-Tuning.** Comparison of CEN-EndoViT (CEN) and FL-EndoViT (FL) across all tasks. Superior models are bolded; (*) denotes statistical significance ($p < 0.01$). Values are mean $\pm$ std (%).

| Task / Metric | Full Fine-Tuning | | Frozen Backbone | |
|---|---|---|---|---|
| | **CEN** | **FL** | **CEN** | **FL** |
| *Surgical Scene Segmentation (SSS) – IoU* | | | | |
| High Res | **67.81 $\pm$ 8.03**$^*$ | 65.98 $\pm$ 8.10 | **68.57 $\pm$ 8.14**$^*$ | 67.23 $\pm$ 7.37 |
| Low Res | **67.54 $\pm$ 8.06**$^*$ | 67.42 $\pm$ 8.18 | **67.28 $\pm$ 8.94**$^*$ | 65.22 $\pm$ 7.55 |
| *Action Triplet Recognition (ATR) – mAP* | | | | |
| | 31.33 $\pm$ 4.61 | **40.79 $\pm$ 6.65**$^*$ | **31.33 $\pm$ 4.61**$^*$ | 26.90 $\pm$ 5.58 |
| *Surgical Phase Recognition (SPR) – Accuracy* | | | | |
| Stage 1 | **81.58 $\pm$ 8.02** | 81.46 $\pm$ 7.59 | **72.26 $\pm$ 8.96**$^*$ | 65.72 $\pm$ 10.72 |
| Stage 2 | 88.37 $\pm$ 7.16 | **89.04 $\pm$ 7.03** | **79.88 $\pm$ 10.88**$^*$ | 75.26 $\pm$ 11.55 |
| *Surgical Phase Recognition (SPR) – F1 Score* | | | | |
| Stage 1 | **71.47 $\pm$ 8.46** | 71.19 $\pm$ 8.26 | **59.71 $\pm$ 8.70**$^*$ | 53.39 $\pm$ 8.82 |
| Stage 2 | 84.48 $\pm$ 6.15 | **85.05 $\pm$ 5.92** | **75.12 $\pm$ 9.50**$^*$ | 72.29 $\pm$ 9.46 |

An ablation study was conducted on three tasks SSS, ATR, and SPR to evaluate whether full fine-tuning of the EndoViT model is required, or if updating only the task-specific head while keeping the backbone frozen is sufficient. We trained models on the full dataset and

---

**Algorithm 1** `SAM/ASAM` and `SWA` applied to `FedAvg` (adapted from (Caldarola et al., 2022))

---

**Input:** Initial random model $f_\theta^0$, $K$ clients, $T$ rounds, learning rates $\gamma_1, \gamma_2$, neighborhood size $\rho > 0$, $\eta > 0$, batch size $|\mathcal{B}|$, local epochs $E$, cycle length $c$

**for** *each round* $t = 0$ **to** $T - 1$ **do**

    **if** $t < 0.75T$ **then**

        $\theta_{\texttt{SWA}} \leftarrow \theta^t$                                                   `// Initialize SWA`

    **end**

    **if** $t \geq 0.75T$ **then**

        $\gamma = \gamma(t)$                                                 `// Compute LR for the round`

    **end**

    Subsample a set $\mathcal{C}$ of clients

    **for** *each client* $k$ *in* $\mathcal{C}$ *in parallel* **do**

        $\theta_{k,0}^{t+1} \leftarrow \theta^t$

        **for** $e = 0$ **to** $E - 1$ **do**

            **for** $i = 0$ **to** $N_k/|\mathcal{B}|$ **do**

                Compute gradient $\nabla_\theta \mathcal{L}_\mathcal{B}(\theta_{k,i}^{t+1})$ on batch $\mathcal{B}$ from $\mathcal{D}_k$

                Compute $\hat{\epsilon} = \rho \dfrac{\nabla_\theta \mathcal{L}_\mathcal{B}(\theta_{k,i}^{t+1})}{\|\nabla_\theta \mathcal{L}_\mathcal{B}(\theta_{k,i}^{t+1})\|_2}$

                $\theta_{k,i+1}^{t+1} \leftarrow \theta_{k,i}^{t+1} - \gamma \left( \left. \nabla_\theta \mathcal{L}_\mathcal{B}(\theta_{k,i}^{t+1}) \right|_{\theta + \hat{\epsilon}} \right)$

           **end**

        **end**

        Send $\theta_k^{t+1}$ to the server

    **end**

    $\theta^{t+1} \leftarrow \sum_{k \in \mathcal{C}} \dfrac{N_k}{\sum_{j \in \mathcal{C}} N_j} \theta_k^{t+1}$                                  `// FedAvg`

    **if** $t \geq 0.75T$ **and** $t \pmod c = 0$ **then**

        $n_{\text{models}} \leftarrow t/c$    $\theta_{\texttt{SWA}} \leftarrow \dfrac{\theta_{\texttt{SWA}} \cdot n_{\text{models}} + \theta^{t+1}}{n_{\text{models}} + 1}$             `// Update SWA`

    **end**

**end**

---

compared the performance of fully fine-tuned models against those with a frozen EndoViT backbone. The performance differences are summarized in Table 6.

Our results indicate that freezing the EndoViT backbone and fine-tuning only the task-specific head generally leads to suboptimal performance across most tasks. The degradation is particularly severe when FL-EndoViT is used, especially for ATR and SPR, where full fine-tuning yields substantial improvements, up to 13.89% for ATR, 17.80 % for SPR-S1 (F1), and 12.76% for SPR-S2 (F1). For CEN-EndoViT, the performance gap is less pronounced but still noticeable. Interestingly, in specific cases such as SSS-High Res, freezing the backbone even results in slight performance improvement, suggesting that important pretrained representations are lost when the backbone is not remained fixed. Neverthe-

less, the observed gap in centralized and federated settings, full fine-tuning is beneficial for extracting task-specific representations.

Given these findings, we focus exclusively on the fully fine-tuned experiments in the subsequent sections. Full fine-tuning outperforms the frozen-backbone approach across nearly all tasks, particularly for FL-EndoViT where freezing the backbone results in significant performance loss. By concentrating on fully fine-tuned models, we ensure that our analysis reflects the most effective approach for leveraging the EndoViT backbone in SDS applications.

## Appendix C. Detailed Results of EndoViT Experiments

### C.1. Pretraining

The full pretraining results are reported in Table 7.

Table 7: **Pretraining Performance Comparison:** The distribution of patch count below different thresholds of the reconstruction loss for Surgical Scene Segmentation (SSS), Action Triplet Recognition (ATR), and Surgical Phase Recognition (SPR), shows the importance of combining MAE with adaptive FedSAM to get results that correspond to centralized (CEN) pretraining.

| | Maximum Number of Patches below Threshold | | | | | | | | |
|---|---|---|---|---|---|---|---|---|---|
| **Pretraining for** | **SSS** | | | **ATR** | | | **SPR** | | |
| **Variant** | **CEN** | **Ours** | **MAE** | **CEN** | **Ours** | **MAE** | **CEN** | **Ours** | **MAE** |
| Patches below 0.3 | 171 | 168 | 67 | 172 | 168 | 67 | 172 | 167 | 67 |
| Patches below 0.3 SWA | 171 | 166 | 67 | 172 | 167 | 67 | 172 | 165 | 67 |
| Patches below 0.1 | 126 | 112 | 25 | 127 | 112 | 25 | 127 | 110 | 25 |
| Patches below 0.1 SWA | 120 | 108 | 25 | 121 | 110 | 25 | 124 | 105 | 25 |
| Patches below 0.05 | 92 | 82 | 11 | 96 | 81 | 11 | 96 | 80 | 11 |
| Patches below 0.05 SWA | 90 | 80 | 11 | 91 | 80 | 11 | 92 | 77 | 11 |
| Patches below 0.01 | 45 | 39 | 0 | 45 | 39 | 0 | 46 | 36 | 0 |
| Patches below 0.01 SWA | 45 | 39 | 0 | 46 | 39 | 0 | 46 | 36 | 0 |

### C.2. Downstream Tasks

#### C.2.1. FEW-SHOT VIDEOS

In the few-shot experiments, three technical repetitions were performed, with each run trained on distinct, non-overlapping video datasets. The Table C.2.1 lists the video IDs of the subsections used in each run.

Table 8: Computational cost analysis per federated training round. The table details the local dataset size (Count) and the resulting MACs (Multiply-Accumulate Operations) (Ops) required for one full epoch per client.

| Client | Count | Fraction | Ops ($\times 10^{12}$) |
|---|---|---|---|
| Cholec80_for_Segmentation | 178,129 | 24.25% | 40.40 |
| DSAD | 13,195 | 1.80% | 2.99 |
| ESAD | 43,456 | 5.92% | 9.86 |
| GLENDA_v1.0 | 1,083 | 0.15% | 0.25 |
| HeiCo | 350,539 | 47.72% | 79.51 |
| LapGyn4_v1.2 | 38,192 | 5.20% | 8.66 |
| PSI_AVA | 73,618 | 10.02% | 16.70 |
| SurgicalActions160 | 761 | 0.10% | 0.17 |
| hSDB-instrument | 35,576 | 4.84% | 8.07 |
| **TOTAL** | **734,549** | **100.00%** | **166.61** |

Table 9: Video IDs used in the few-shot experiments for SSS, ATR, and SPR across three technical repetitions. Each run utilizes distinct, non-overlapping video datasets, with varying numbers of videos for each task.

| SSS | 1 Video | 2 Videos | 4 Videos |
|---|---|---|---|
| **Run 1** | 9 | 24, 1 | 18, 48, 25, 1 |
| **Run 2** | 28 | 25, 48 | 28, 55, 24, 37 |
| **Run 3** | 43 | 55, 28 | 37, 18, 24, 20 |
| **ATR** | **2 Videos** | **4 Videos** | **8 Videos** |
| **Run 1** | 27, 5 | 36, 14, 57, 15 | 18, 57, 51, 60, 27, 68, 36, 48 |
| **Run 2** | 70, 47 | 23, 66, 31, 27 | 2, 36, 5, 47, 48, 8, 6, 18 |
| **Run 3** | 51, 50 | 42, 31, 6, 52 | 47, 57, 8, 80, 15, 68, 40, 27 |
| **SPR** | **2 Videos** | **4 Videos** | **8 Videos** |
| **Run 1** | 25, 1 | 8, 28, 13, 4 | 1, 22, 12, 8, 6, 33, 31, 29 |
| **Run 2** | 33, 11 | 5, 17, 25, 31 | 20, 38, 25, 7, 29, 4, 2, 24 |
| **Run 3** | 22, 40 | 24, 28, 38, 15 | 18, 28, 17, 2, 40, 36, 7, 34 |

### C.2.2. FULLY FINE-TUNED EXPERIMENTS

Detailed insights of the fully fine-tuned experiments is available in Figure 2, Figure 3, Figure 4, Figure 5, and Figure 6.

### C.2.3. FROZEN BACKBONE EXPERIMENTS

Detailed insights of the frozen backbone experiments is available in Figure 7, Figure 8, Figure 9, Figure 10, and Figure 11.

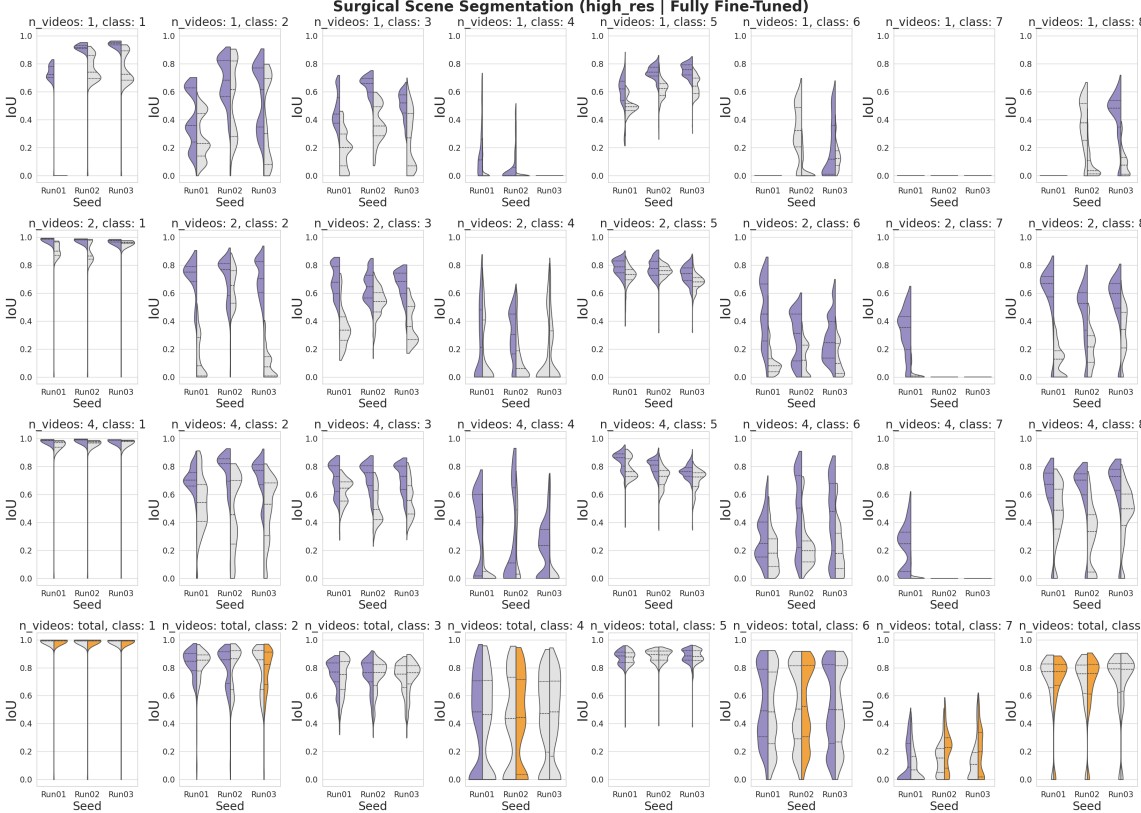

Figure 2: **Performance Comparison of Surgical Scene Segmentation on High-Resolution Images (Fully Fine-Tuned).** The violin plots illustrate the distribution of IoU (Intersection over Union) scores for 1,040 test images. The left half of the violin plots the federated variant, while the right half plots the centralized variant. The split violins are color-coded: purple indicates a significant improvement in performance with the federated backbone model, orange indicates a significant improvement in performance with the centralized model, and gray indicates no significant difference, as measured by a Wilcoxon signed-rank test.

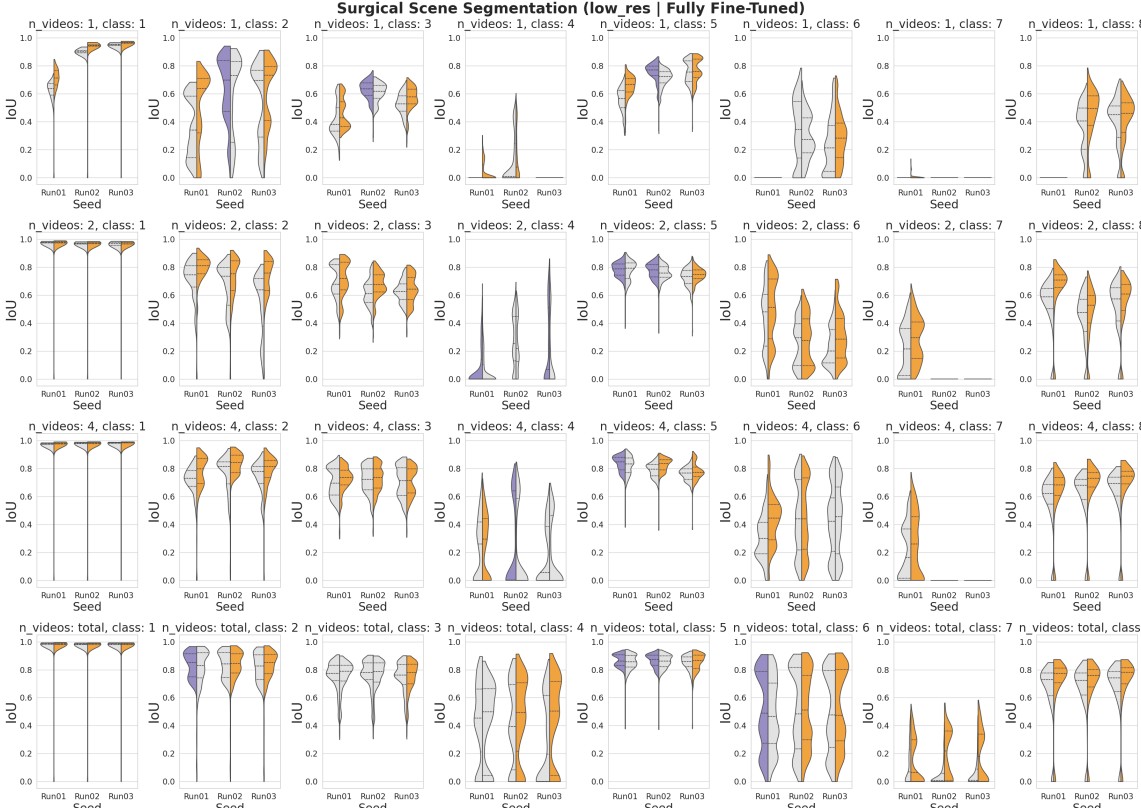

Figure 3: **Performance Comparison of Surgical Scene Segmentation on Low-Resolution Images (Fully Fine-Tuned).** The violin plots illustrate the distribution of IoU (Intersection over Union) scores for 1,040 test images. The left half of the violin plots the federated variant, while the right half plots the centralized variant. The split violins are color-coded: purple indicates a significant improvement in performance with the federated backbone model, orange indicates a significant improvement in performance with the centralized model, and gray indicates no significant difference, as measured by a Wilcoxon signed-rank test.

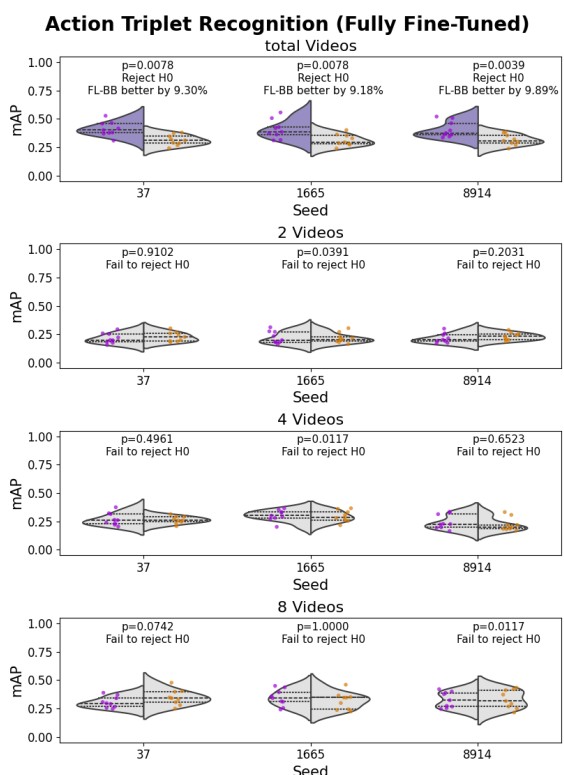

Figure 4: **Performance Comparison of Action Triplet Recognition (Fully Fine-Tuned).** The violin plots show average precision scores distribution for nine test videos. The left half of the violin plots the federated variant, while the right half plots the centralized variant. The split violins are color-coded: purple indicates significantly better performance with the federated backbone model, orange indicates significantly better performance with the centralized model, and gray indicates no significant difference, as measured by a Wilcoxon signed-rank test.

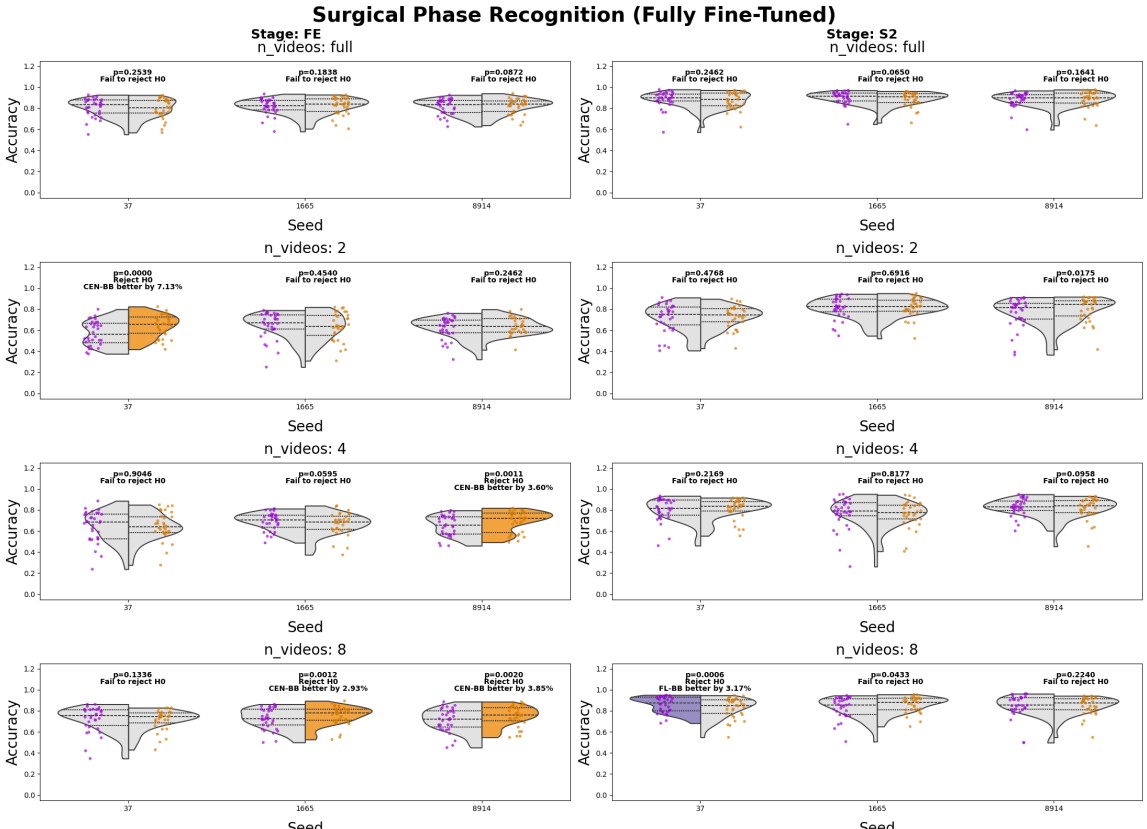

Figure 5: **Performance Comparison of Surgical Phase Recognition (Fully Fine-Tuned, Accuracy).** The violin plots illustrate the accuracy score distributions for the two-stage TeCNO approach across 31 videos. The left subfigure depicts the initial stage, Feature Extraction without temporal dimension, while the right subfigure illustrates the subsequent stage with temporal dimension. The left half of the violin plots the federated variant, and the right half plots the centralized variant. The violin plots are further annotated with color-coded labels: purple signifies significantly superior performance with the federated backbone model, orange indicates significantly superior performance with the centralized model, and gray denotes no significant difference, as determined by a Wilcoxon signed-rank test.

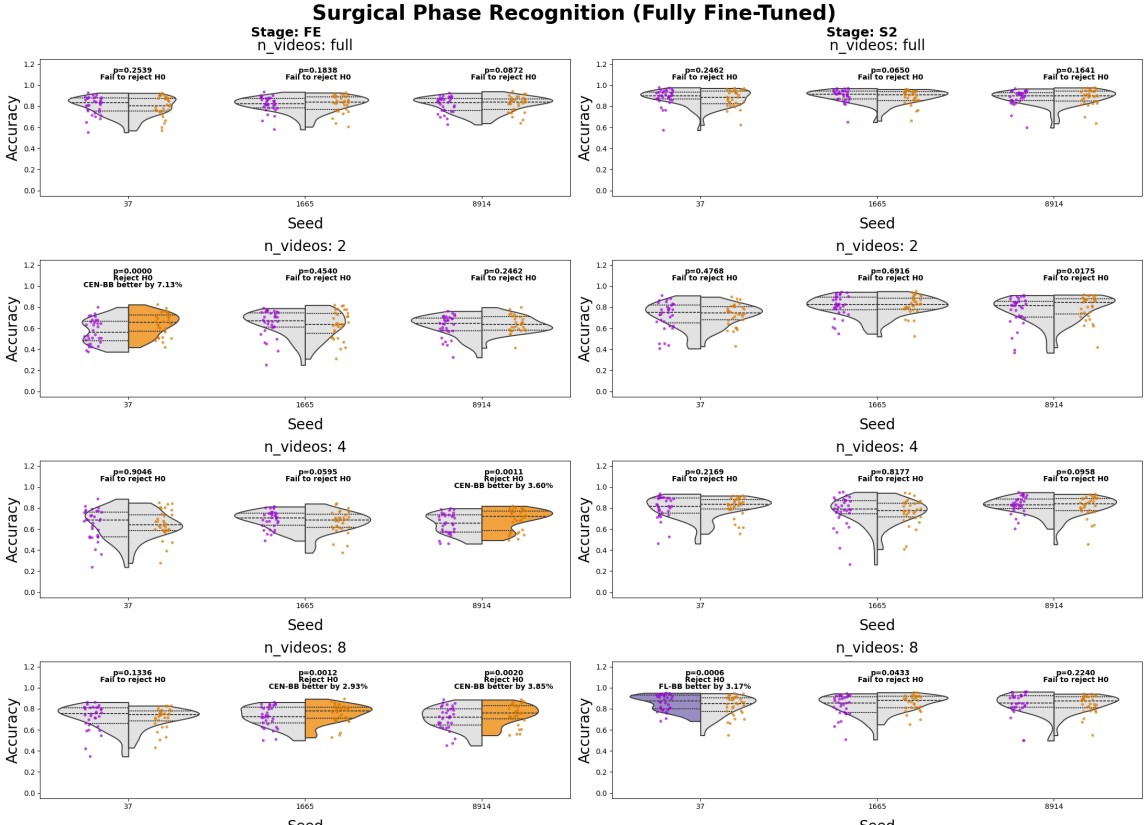

Figure 6: **Performance Comparison of Surgical Phase Recognition (Fully Fine-Tuned, Accuracy).** The violin plots illustrate the accuracy score distributions for the two-stage TeCNO approach across 31 videos. The left subfigure depicts the initial stage, Feature Extraction without temporal dimension, while the right subfigure illustrates the subsequent stage with temporal dimension. The left half of the violin plots the federated variant, and the right half plots the centralized variant. The violin plots are further annotated with color-coded labels: purple signifies significantly superior performance with the federated backbone model, orange indicates significantly superior performance with the centralized model, and gray denotes no significant difference, as determined by a Wilcoxon signed-rank test.

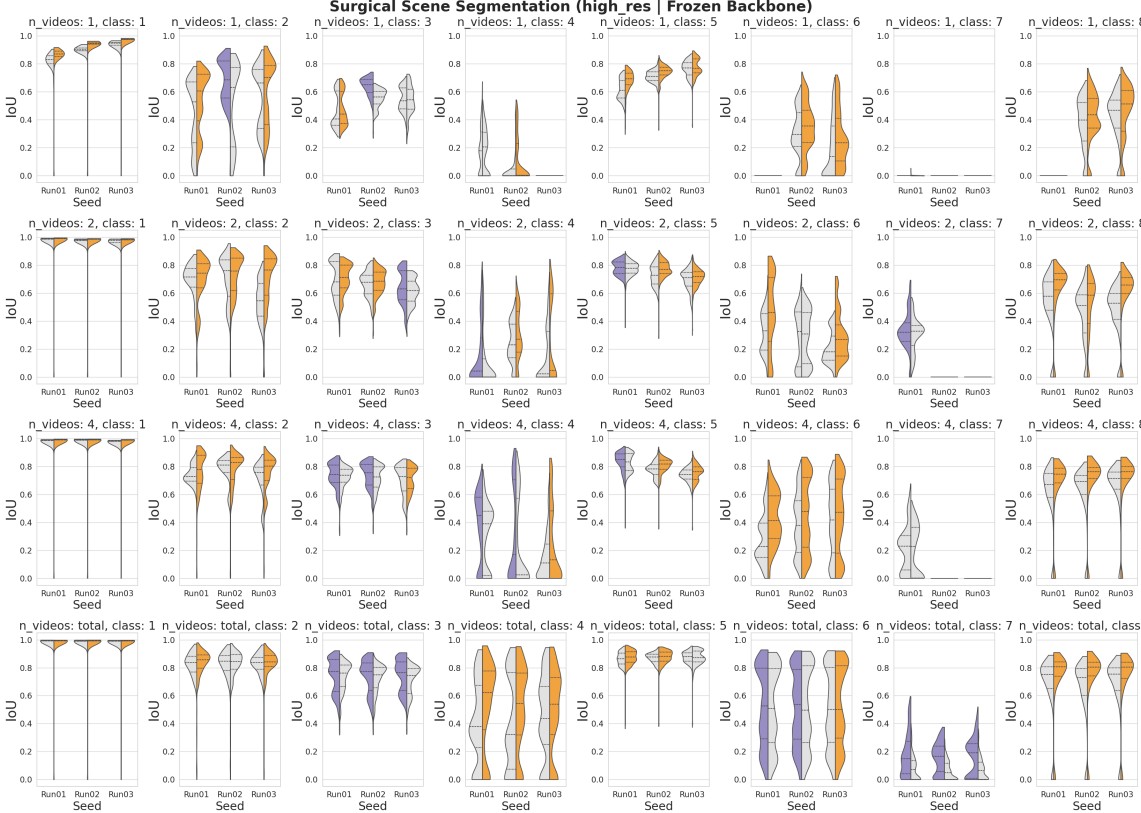

Figure 7: **Performance Comparison of Surgical Scene Segmentation on High-Resolution Images (Frozen Backbone).** The violin plots illustrate the distribution of IoU (Intersection over Union) scores for 1,040 test images. The left half of the violin plots the federated variant, while the right half plots the centralized variant. The split violins are color-coded: purple indicates a significant improvement in performance with the federated backbone model, orange indicates a significant improvement in performance with the centralized model, and gray indicates no significant difference, as measured by a Wilcoxon signed-rank test.

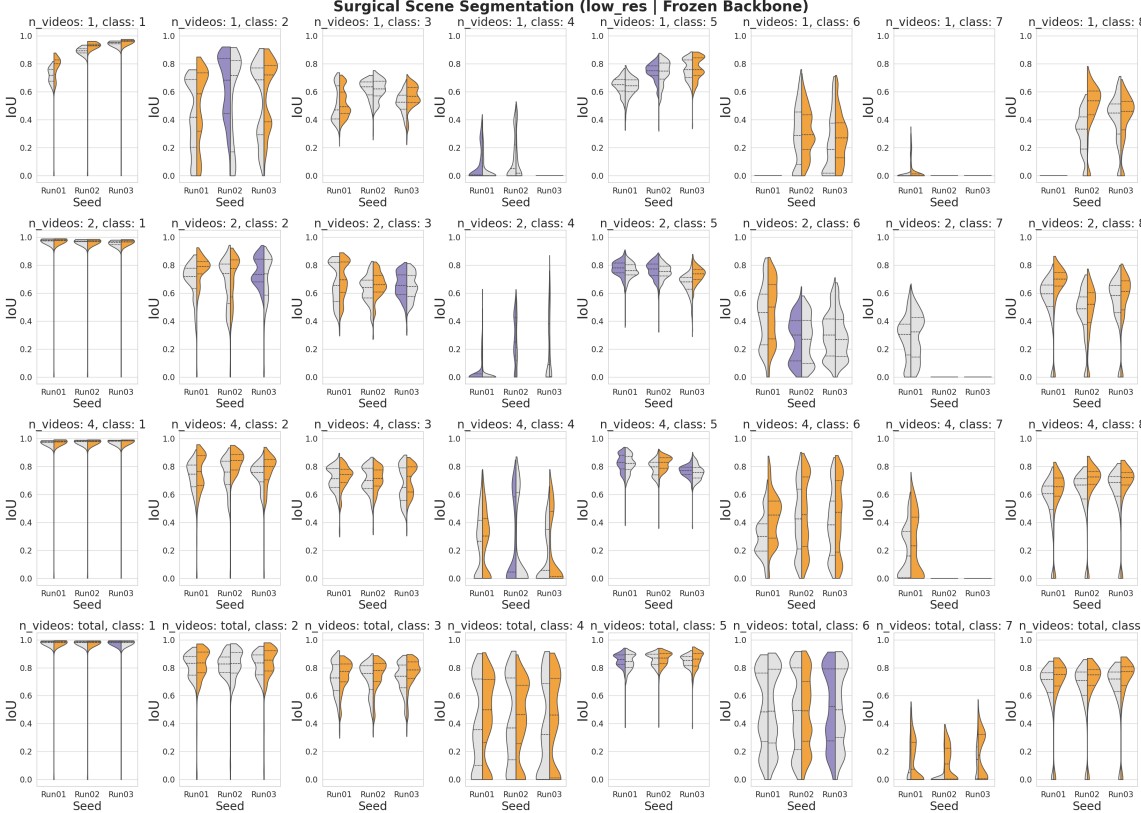

Figure 8: **Performance Comparison of Surgical Scene Segmentation on Low-Resolution Images (Frozen Backbone).** The violin plots illustrate the distribution of IoU (Intersection over Union) scores for 1,040 test images. The left half of the violin plots the federated variant, while the right half plots the centralized variant. The split violins are color-coded: purple indicates a significant improvement in performance with the federated backbone model, orange indicates a significant improvement in performance with the centralized model, and gray indicates no significant difference, as measured by a Wilcoxon signed-rank test.

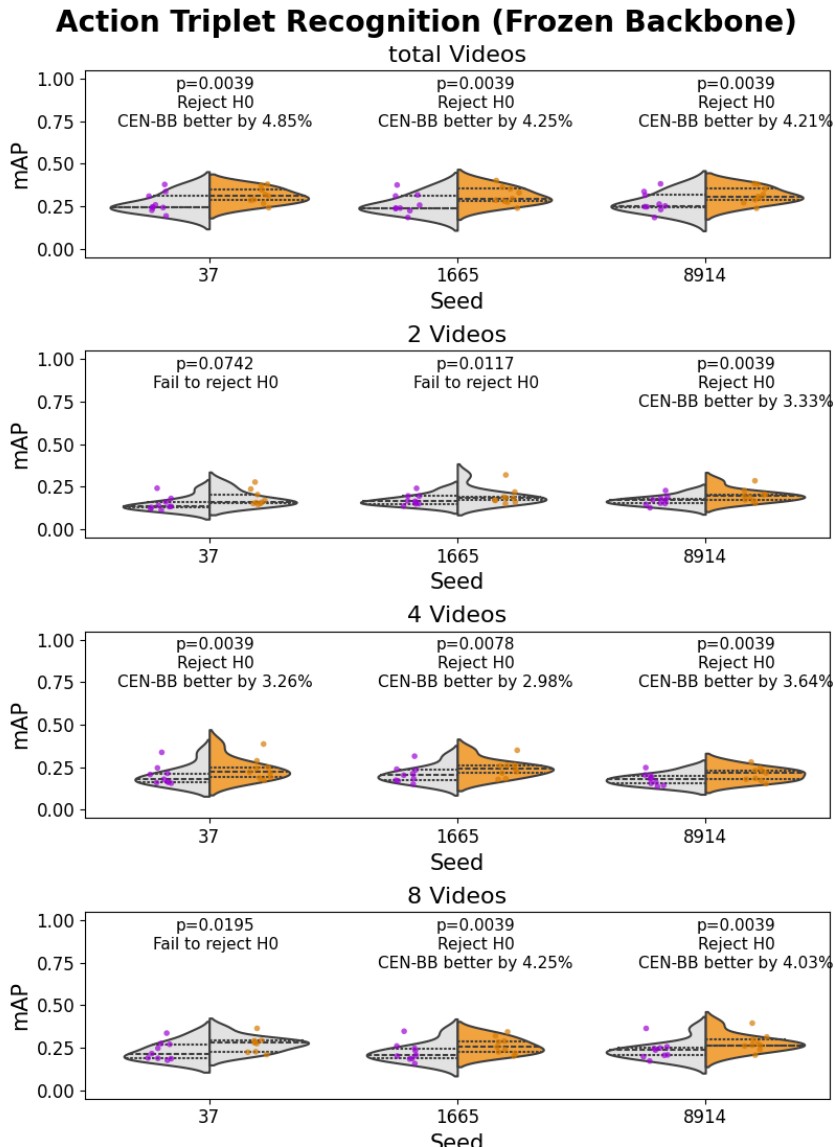

Figure 9: **Performance Comparison of Action Triplet Recognition (Frozen Backbone).** The violin plots show average precision scores distribution for nine test videos. The left half of the violin plots the federated variant, while the right half plots the centralized variant. The split violins are color-coded: purple indicates significantly better performance with the federated backbone model, orange indicates significantly better performance with the centralized model, and gray indicates no significant difference, as measured by a Wilcoxon signed-rank test.

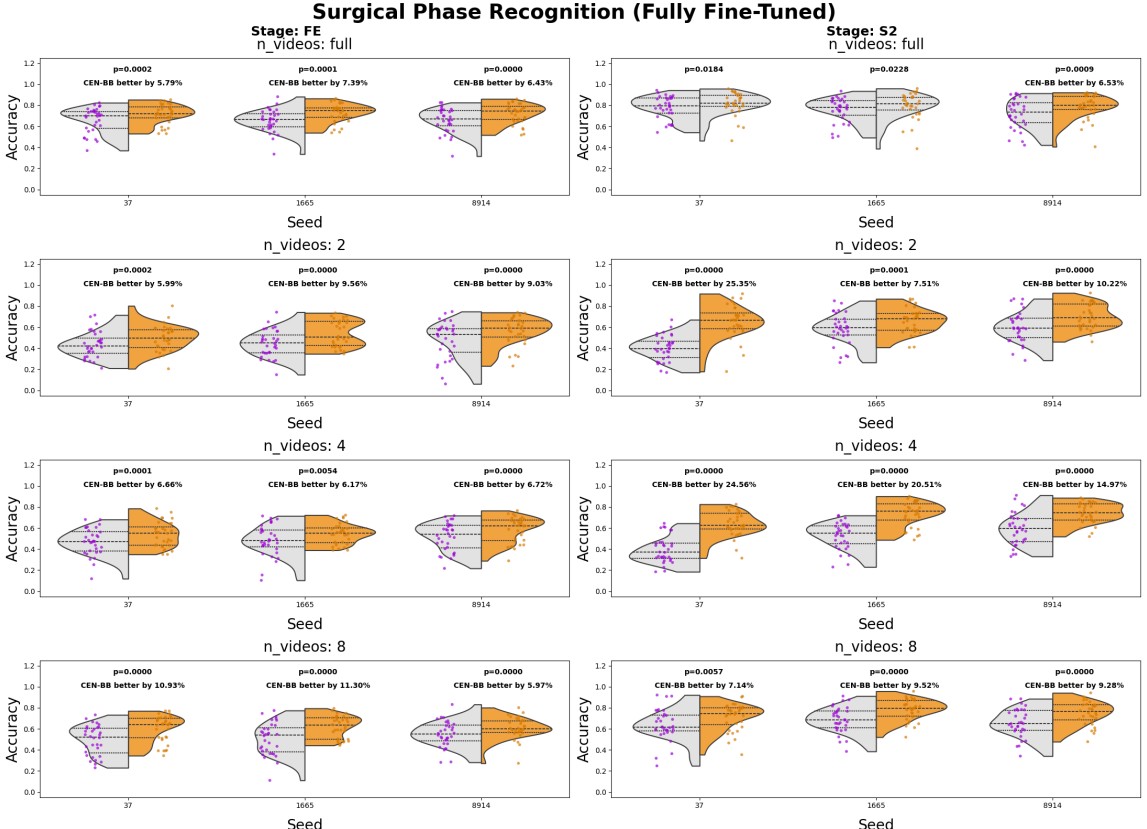

Figure 10: **Performance Comparison of Surgical Phase Recognition (Frozen Backbone, Accuracy).** The violin plots illustrate the accuracy score distributions for the two-stage TeCNO approach across 31 videos. The left subfigure depicts the initial stage, Feature Extraction without temporal dimension, while the right subfigure illustrates the subsequent stage with temporal dimension. The left half of the violin plots the federated variant, and the right half plots the centralized variant. The violin plots are further annotated with color-coded labels: purple signifies significantly superior performance with the federated backbone model, orange indicates significantly superior performance with the centralized model, and gray denotes no significant difference, as determined by a Wilcoxon signed-rank test.

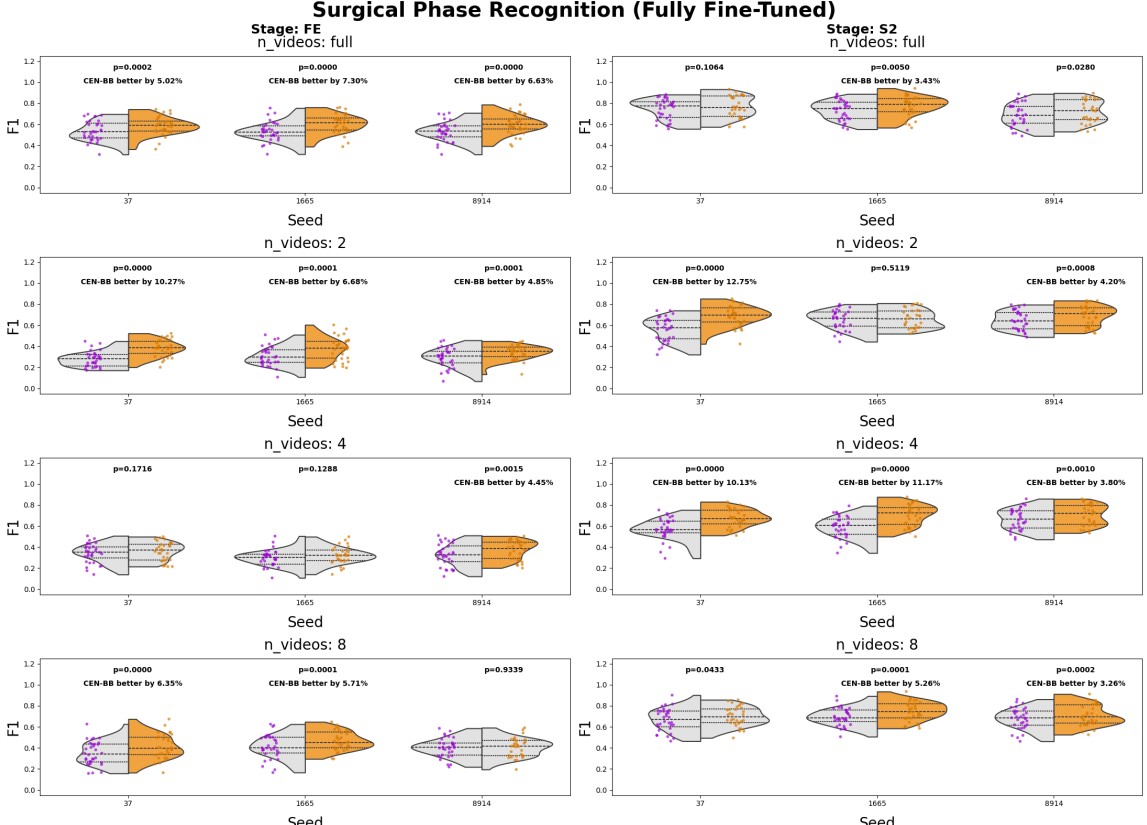

Figure 11: **Performance Comparison of Surgical Phase Recognition (Frozen Backbone, Accuracy).** The violin plots illustrate the accuracy score distributions for the two-stage TeCNO approach across 31 videos. The left subfigure depicts the initial stage, Feature Extraction without temporal dimension, while the right subfigure illustrates the subsequent stage with temporal dimension. The left half of the violin plots the federated variant, and the right half plots the centralized variant. The violin plots are further annotated with color-coded labels: purple signifies significantly superior performance with the federated backbone model, orange indicates significantly superior performance with the centralized model, and gray denotes no significant difference, as determined by a Wilcoxon signed-rank test.

## Appendix D. GynSurg Setup and Results

### D.1. Dataset and Tasks

The GynSurg dataset (Nasirihaghighi et al., 2025) comprises three video classification sub-tasks:

1. **Bleeding and Smoke Recognition:** Two separate binary classification tasks.

2. **Action Recognition:** A multi-class classification task with five classes: Needle Passing, Coagulation, Suction and Irrigation, Transection, and Rest.

### D.2. Methodology and Evaluation

Following the experimental setup of the original GynSurg paper (Nasirihaghighi et al., 2025), we extracted features from our pretrained backbones and fed them into a two-layer stacked LSTM with a hidden size of 256. The output of the final LSTM layer served as the embedding for classification. Performance was evaluated using a 4-fold cross-validation protocol. Statistical significance between the FL-EndoViT and CEN-EndoViT backbones was determined using a paired Wilcoxon signed-rank test on the fold-level results. Table 10 presents the detailed per-fold and mean results for all GynSurg subtasks and Table 11 presents the class-wise comparison for the action recognition task between FL-EndoViT and CEN-EndoViT.

Table 10: GynSurg 4-fold cross-validation results. **RN50**: ResNet50+LSTM, **FL**: FL-EndoViT+LSTM, **CEN**: CEN-EndoViT+LSTM. Values in %. Best results bolded.

| Task | Fold | RN50 | | FL | | CEN | |
|---|---|---|---|---|---|---|---|
| | | **Acc** | **F1** | **Acc** | **F1** | **Acc** | **F1** |
| **Action** | 1 | 69.72 | 59.03 | 63.88 | 57.78 | 64.99 | 57.66 |
| | 2 | 76.48 | 70.53 | 64.66 | 56.01 | 68.45 | 61.28 |
| | 3 | 80.58 | 75.41 | 73.66 | 71.48 | 79.67 | 74.70 |
| | 4 | 77.52 | 76.62 | 74.37 | 74.66 | 63.24 | 64.07 |
| | *Avg* | **76.1±4.6** | **70.4±8.0** | 69.1±5.6 | 65.0±9.5 | 69.1±7.4 | 64.4±7.3 |
| **Bleed** | 1 | 87.12 | 85.68 | **88.77** | **87.25** | 85.48 | 83.65 |
| | 2 | **93.21** | **92.97** | 89.30 | 88.68 | 63.58 | 50.62 |
| | 3 | **91.15** | **90.89** | 80.71 | 79.40 | 88.11 | 87.86 |
| | 4 | **96.77** | **96.71** | 71.13 | 71.07 | 75.29 | 73.84 |
| | *Avg* | **92.1±4.0** | **91.6±4.6** | 82.5±8.5 | 81.6±8.1 | 78.1±11.2 | 74.0±16.7 |
| **Smoke** | 1 | 79.00 | 79.00 | **84.73** | **84.55** | 78.20 | 78.20 |
| | 2 | 86.29 | 86.27 | **87.03** | **87.01** | 87.03 | 86.96 |
| | 3 | 79.86 | 79.83 | 79.44 | 79.38 | **85.49** | **85.45** |
| | 4 | 93.08 | 90.72 | **96.12** | **94.74** | 95.02 | 93.23 |
| | *Avg* | 84.6±6.6 | 84.0±5.6 | **86.8±7.0** | **86.4±6.4** | 86.4±6.9 | 86.0±6.2 |

Table 11: Class-wise comparison of the GynSurg action recognition tasks (all folds combined). **CEN**: CEN-EndoViT, **FL**: FL-EndoViT. All values are reported in percentage (%).

| Class | F1-Score | | | Accuracy | | |
|---|---|---|---|---|---|---|
| | CEN | FL | Diff | CEN | FL | Diff |
| 0 - Coagulation | 66.10 | 59.70 | 6.40 | 57.20 | 45.70 | 11.50 |
| 1 - Needle Passing | 83.90 | 88.40 | -4.50 | 76.60 | 86.10 | -9.50 |
| 2 - Rest | 65.50 | 62.20 | 3.30 | 82.50 | 79.10 | 3.40 |
| 3 - Suction and Irrigation | 42.80 | 50.00 | -7.20 | 41.50 | 55.20 | -13.70 |
| 4 - Transection | 63.80 | 68.90 | -5.10 | 58.20 | 61.60 | -3.40 |

## Appendix E. Machine Learning Setup

Pretraining in simulation mode was carried out in parallel on four NVIDIA V100 GPUs that do not share memory, while the centralized baseline was retrained on a single NVIDIA V100 GPU with 28 CPUs per task. Centralized fine-tuning for SSS, ATR, SPR, and GynSurg was performed on a single NVIDIA V100 GPU with 12 CPUs. For all experiments we used PyTorch 1.13.0.

