# OpenReview forum: "Federated EndoViT: Pretraining Vision Transformers via Federated Learning on Endoscopic Image Collections"
_MIDL.io/2026/Conference — MIDL 2026 Poster_

### Official Review · Reviewer_8x8v · 2026-01-05

**Confidence:** 1
**Preliminary Rating:** 3
**Final Rating:** 4

**Summary:**

This paper trains an endoscopy Vision Transformer using federated learning, so clients can pretrain a strong model without sharing raw data. They use MAE self-supervised pretraining and show that standard federated learning doesn’t work well, but FedSAM + server-side SWA makes federated MAE pretraining stable and close to centralized training. They then fine-tune the pretrained model on several surgical tasks and find that the FL-pretrained model is often comparable to centralized and sometimes better.

**Strengths:**

Normal federated learning methods don’t work well for MAE-style pretraining in this endoscopy setting, but the paper shows that using FedSAM plus server-side SWA makes federated pretraining stable and close to centralized training.

They evaluate on many downstream tasks, which shows the generalizability of their method.

**Weaknesses:**

They mention substantial communication overhead as a practical challenge, but don’t report the quantitative difference.

The benefits are not consistent across tasks/settings. In some cases, centralized training is stronger or more stable

**Detailed Comments:**

"In example in the SSS task the federated model performance is slightly lower in low-resolution settings, indicating that the pre-training strategy is not as effective as centralized learning when resolution decreases and data heterogeneity exists. " can you use several sentences to state the reason?

**Justification Of Final Rating:**

The authors have carefully addressed all of my concerns in their response. The added clarifications and explanations resolve the previously raised issues and improve the overall clarity of the paper. The response sufficiently strengthens the submission, so I update my evaluation to a weak accept.

**Justification Of The Preliminary Rating:**

I have a limited background in federated learning, so my comments focus mainly on the experimental evidence and overall methodology rather than on the novelty or technical details of the federated learning components.

**Questions To Address In The Rebuttal:**

Can you show the quantitative results in computational resources or communication cost, and compare them to centralized training?

---

> ### Author Response · Authors · 2026-01-23
>
> We thank the reviewer for their constructive feedback and for identifying the need to clarify performance nuances in low-resolution settings.
>
> **Q1: Can you show the quantitative results in computational resources or communication cost, and compare them to centralized training?**
> A: This important point regarding resource efficiency was also noted by Reviewer n3Z5. Our analysis indicates that while the computational load (166.6 T-MACs per epoch) remains constant regardless of the setting, the federated approach introduces specific communication and distribution challenges. The total data transfer over 15 rounds is approximately 121 GB (plus protocol overhead), which is higher than the 82 GB required for centralized aggregation but ensures data sovereignty. Furthermore, the computational load is heavily skewed due to dataset imbalance, with the largest site (HeiCo) performing 47.7% of the operations. We have added a section to the Discussion. Furthermore, we recommended hardware configuration, including 32GB VRAM GPUs and 1 Gbps network connections, for clinical translation purposes to support the demands of such an heterogeneous federated network.
>
>
> **Q2: "In example in the SSS task the federated model performance is slightly lower in low-resolution settings, indicating that the pre-training strategy is not as effective as centralized learning when resolution decreases and data heterogeneity exists." Can you use several sentences to state the reason?**
>
> A: We attribute this to the averaging nature of Federated Learning. During pre-training, the aggregation of model updates creates a strong general representation but tends to smooth out fine-grained details, such as sharp object boundaries. When fine-tuning with high-resolution data, the model receives enough detailed signal to "recover" these sharp edges. However, low-resolution data lacks the pixel density required to refine these smoothed features. Consequently, the federated model cannot fully recover the precise edge details, whereas the centralized model retains them from the start by accessing raw data. We have revised the Discussion section to explicitly include this analysis regarding resolution-dependent performance.

---

### Official Review · Reviewer_F6CG · 2026-01-09

**Confidence:** 5
**Preliminary Rating:** 4
**Final Rating:** 4

**Summary:**

This work studies how privacy regulations limit data sharing across hospitals and proposes Federated EndoViT (FL-EndoViT) as a solution for training surgical foundation models without exchanging raw data. The method adapts masked autoencoder pretraining to federated learning and uses the adaptive optimizer FedSAM to handle strong differences between institutions. FL-EndoViT is pretrained on the large Endo700k dataset and evaluated on scene segmentation, action recognition, and surgical phase recognition. The federated model achieves performance comparable to centralized training, with clear benefits in data-scarce and high-resolution segmentation and improved generalization to unseen surgical events. Overall, the results show that adaptive federated learning is a practical path toward robust and privacy-preserving surgical foundation models.

**Strengths:**

The paper has several clear strengths. Most importantly, the evaluation is robust, cohesive, and well designed, covering multiple relevant downstream tasks and including meaningful comparisons to centralized training, which convincingly supports the claims. The writing, structure, and overall presentation are strong, with a clear problem motivation, a logical flow, and an effective overview figure that helps readers quickly grasp the method and contributions. The work is well grounded in prior literature, situating itself appropriately within federated learning, surgical data science, and foundation models, and it follows sound scientific principles with thorough ablations and analysis.

While the core method can be seen largely as a reapplication and careful integration of existing federated learning and optimization ideas to the EndoViT setting rather than a fundamentally new algorithmic contribution, the paper’s value lies in demonstrating that these techniques work reliably in a challenging and practically important domain. This makes the work highly relevant to the community delivery key insights, especially given real-world privacy constraints in surgery.

**Weaknesses:**

The main weakness of the paper is that the methodological novelty is limited, as the core contribution largely consists of applying Sharpness-Aware Minimization in a federated setting to EndoViT. While this adaptation is carefully engineered and well validated, similar optimization strategies for handling heterogeneity in federated learning have already been studied in the broader FL literature, for example variance-reduction and sharpness-aware approaches that explicitly target generalization under non-IID data, such as the CVPR 2023 work by Li et al. on partial variance reduction. As a result, the paper’s contribution is better framed as a strong validation and domain transfer rather than a new federated learning method.

A second limitation concerns the centralized baseline and data scaling claims. Constructing a centralized model trained on nine datasets serves as a useful upper bound, but it represents an idealized scenario that is rarely achievable in real clinical practice. While the paper motivates federated learning as a way to access more data without pooling, it does not explicitly demonstrate that the proposed FL setup benefits from increasing data diversity or scale, for example by incrementally adding more heterogeneous surgical domains beyond the Endo700k training data. Since data quality and domain relevance are critical for foundation model performance, it remains unclear whether FL-EndoViT would continue to improve over the centralized version when exposed to more diverse or loosely related surgical videos, or whether negative transfer might occur.

Finally, although the paper shows that FL can match centralized performance, it does not engage with more advanced federated learning algorithms or leverages it core advantages, which can sometimes outperform centralized training in terms of generalization under heterogeneity, particularly when more dynamic or adaptive aggregation strategies are used Babendererde, et al. 2025. A more explicit comparison or discussion would strengthen the claims and better position the method relative to existing FL advances.

Li, Bo, et al. "On the effectiveness of partial variance reduction in federated learning with heterogeneous data." Proceedings of the IEEE/CVF conference on computer vision and pattern recognition. 2023.

Babendererde, Niklas, et al. "Federated-continual dynamic segmentation of histopathology guided by barlow continuity." 2025 IEEE/CVF Winter Conference on Applications of Computer Vision (WACV). IEEE, 2025.

**Detailed Comments:**

In Figure 1, the server-side box lists both Aggregation and SWA, although SWA is itself a form of model aggregation. This may be confusing to readers. The figure could be improved by either merging these into a single aggregation block or explicitly labeling SWA as a specific aggregation strategy. If no additional server-side mechanisms such as distillation are used, it may also be helpful to state this explicitly or visually simplify the server module to better reflect the actual pipeline used in the paper.

In Table 1, the optimizer is referred to as FedASAM, whereas the rest of the paper consistently uses FedSAM. This appears to be a typographical inconsistency and should be corrected to avoid confusion.

The citation to Anonymous. FEDKIM should be updated, as this work has since been published. Replacing the anonymous reference with the final published version would improve clarity and ensure proper attribution.

**Justification Of Final Rating:**

Thanks for the answers to the authors.

The rebuttal addresses the main concerns in a clear and technically sound manner, particularly by clarifying the optimization choices, the role of FedSAM in stabilizing federated MAE pretraining, and the deliberate scope limitations regarding dynamic aggregation and cross-domain scaling. These clarifications strengthen the paper’s positioning as a careful and non-trivial validation study in a highly challenging federated surgical setting. The authors also appropriately acknowledge limitations and outline realistic future directions, which improves transparency and scientific rigor.

**Justification Of The Preliminary Rating:**

The paper addresses an important problem in surgical data science and is clearly written, well structured, and carefully evaluated across multiple relevant tasks. The experiments convincingly show that federated pretraining can match centralized performance under realistic privacy constraints, and the work is well positioned within existing literature.

However, the methodological contribution is limited, as the approach mainly applies existing federated sharpness-aware optimization techniques to EndoViT rather than introducing a new learning paradigm. Some motivating claims, such as improved scaling with data diversity or advantages over centralized training, are not fully explored. Overall, the paper provides solid validation and practical insight, with impact driven more by evaluation quality than novelty.

**Questions To Address In The Rebuttal:**

How do you explain the extremely large gap between FedSAM and other federated methods? Did the author spend the same amount of time on hyperparameter tuning?

Does FL-EndoViT benefit from increasing data scale and heterogeneity beyond the Endo700k dataset, for example by incrementally adding more diverse surgical domains, and how does the method handle potential negative transfer when data quality or domain relevance varies?

---

> ### Author Response · Authors · 2026-01-23
>
> We thank the reviewer for recognizing the study's value and validating our methodology.
>
> **Q1: Performance & Dynamic Aggregation Performance Gap:** To ensure the disparity reflects intrinsic algorithmic limitations rather than suboptimal configuration, we maintained consistent base hyperparameters (e.g., LR) across all experiments while tuning algorithm-specific parameters for baselines (e.g., Q for QFedAvg) using the Flower framework. The observed gap stems from the sensitivity of SSL with MAE to non-IID medical data, as pixel-level reconstruction tasks are particularly prone to domain shifts. Standard methods typically converge to sharp local minima that are incompatible when averaged, whereas FedSAM explicitly targets flat minima to ensure the global model remains robust during aggregation. We have updated the manuscript to explicitly mention our strategy.
>
> Dynamic Aggregation: We agree that dynamic aggregation strategies, such as the one proposed by Babendererde et al., offer a promising direction for handling heterogeneity. In the present study, however, our primary objective was to evaluate the effect of pretraining via FedSAM, rather than to jointly optimize multiple FL components. Incorporating dynamic aggregation would have introduced additional algorithmic degrees of freedom, making it more difficult to attribute performance gains specifically to the proposed pretraining strategy. Nevertheless, we have updated our Discussion to acknowledge that future iterations could leverage such dynamic aggregation strategies to further boost generalization, potentially surpassing centralized baselines in our highly heterogeneous settings where client drift and catastrophic forgetting are common.
>
> **Q2: Scaling & Negative Transfer Scaling:** We recognize that FL-EndoViT would likely benefit from scaling across diverse surgical domains beyond our current dataset, such as orthopedic or cardiothoracic procedures. Nevertheless, our study's design centers on MIS with a specific emphasis on visceral and similar abdominal procedures. Extending the training data to substantially different fields introduces a higher risk of negative transfer due to differences in anatomy, instrumentation and visual context. Explicitly modeling and mitigating such effects, e.g. through domain-aware aggregation, client weighting or inclusion of new data sources, would require additional components that fall outside the scope of this work. Future work could certainly explore the transferability of FL-EndoViT to these other surgical fields, but they are excluded from the current analysis to maintain a controlled setting that allows us to isolate the benefits of federated pretraining without introducing uncontrolled domain shifts.
> Regarding negative transfer, our method aims to mitigate this effect through the integration of the FedSAM optimizer and SWA. By specifically targeting flatter regions of the loss landscape, FedSAM promotes a global model that is more resilient to the sharp local minima often associated with non-IID or noisy updates. This approach helps reduce the susceptibility to performance degradation when participating centers possess varying levels of domain relevance.
> While the current simulation assumes a baseline of data quality, we have added a dedicated section to our discussion outlining how future translational iterations could further safeguard against negative transfer by incorporating robust aggregation techniques or personalized FL layers to selectively weigh institutional contributions based on their alignment with the global task.
>
> We have added both points (Q1 & Q2) to the discussion of the revised manuscript.
>
> **Q3: Figure 1, Typos, and Citations:** We have updated Figure 1 to clearly represent the server pipeline, corrected the FedSAM typo, and updated the FedKim citation.
>
> **Q4: Contribution Framing:** We agree that our core contribution is not the introduction of a new FL objective in the theoretical sense. Rather, the novelty lies in demonstrating how sharpness-aware optimization can be worked in a highly challenging federated surgical setting, which we believe constitutes a non-trivial contribution in its own right. We wish to highlight that strong validation holds particular value in medical AI where ensuring robustness against severe non-IID shifts such as varying instrumentation and anatomy is essential for clinical applicability.
> Our results in Table 1 indicate this was not a straightforward domain transfer. In fact, standard federated baselines (e.g., FedAvg) fail to converge when applied to EndoViT, underscoring that naïve application of existing methods is insufficient. Enabling stable training required specific adaptations to the optimization strategy, without which ViT are ineffective in this federated context.  We have revised the paper to better articulate that our contribution is the specific adaptation and rigorous validation to enable ViT in this challenging federated setting.

---

### Official Review · Reviewer_n3Z5 · 2026-01-12

**Confidence:** 4
**Preliminary Rating:** 4
**Final Rating:** 5

**Summary:**

This paper introduces FL-EndoViT, a federated learning framework for pretraining surgical foundation models using a Masked Autoencoder (MAE) based Vision Transformer on the large-scale Endo700k dataset. The authors propose integrating Adaptive Federated Sharpness-Aware Minimization (FedSAM) and Stochastic Weight Averaging (SWA) to address severe non-IID data heterogeneity across institutions. Extensive evaluation on Surgical Scene Segmentation (SSS), Action Triplet Recognition (ATR), Surgical Phase Recognition (SPR), and GynSurg video classification demonstrates that FL-EndoViT achieves performance comparable to centralized training and significantly outperforms it in high-resolution few-shot segmentation and generalization to rare events such as bleeding and smoke (e.g., +16.35% mIoU improvement for single-video SSS; higher F1 for bleeding and smoke recognition). The work establishes federated MAE pretraining as a viable path toward privacy-preserving, generalizable surgical foundation models.

**Strengths:**

1. *High scientific and clinical relevance:* The paper directly addresses the core barrier in surgical AI—privacy-restricted data aggregation—by combining foundation models with federated learning, an important and timely problem in Surgical Data Science.
2. *Strong methodological contribution:* The integration of MAE with FedSAM and SWA is technically well-motivated and empirically validated, with convincing ablation results showing that FedSAM is essential for convergence under extreme non-IID distributions (Table 1, Table 7).
3. *Extensive experimental validation:* Evaluation spans multiple datasets and tasks (SSS, ATR, SPR, GynSurg), including few-shot, high/low resolution, and cross-domain generalization, supported by statistical testing (Wilcoxon, α=0.01).
4. *Clear performance benefits:* FL-EndoViT significantly outperforms centralized baselines in challenging settings (e.g., high-resolution SSS few-shot: 42.03% vs 25.68% mIoU; bleeding detection: 82.48% accuracy, 81.60% F1).
5. *Reproducibility and transparency:* Public release of training code enhances credibility and community impact.

**Weaknesses:**

1. Resource intensity:* The framework requires full end-to-end fine-tuning for optimal performance, making training computationally expensive and potentially limiting accessibility for smaller institutions.
2. *Communication overhead and deployment complexity:* While discussed qualitatively, the paper does not provide quantitative analysis of federated communication cost, system scalability, or wall-clock training time, which are important for real-world adoption.
3. *Limited prospective validation:* The study is retrospective and simulation-based; real-time federated deployment challenges such as client dropouts, asynchronous updates, and adversarial robustness are not experimentally evaluated.
4. *Task-dependent performance trade-offs:* In low-resolution segmentation and majority-class action recognition, centralized models still outperform the federated approach, indicating areas where FL representations may oversmooth fine-grained features.

**Detailed Comments:**

1. Providing training time, communication volume, and hardware requirements would strengthen the practical relevance of the work.
2. A brief analysis of robustness to client imbalance and dropouts would improve confidence in real-world deployment.
3. Future work could explore video-based federated pretraining, as suggested, to better exploit temporal dynamics in surgery.
4. The clarity of the figures (e.g., Figures 2–5) is high and effectively communicates comparative behavior across tasks.

**Justification Of Final Rating:**

Authors fully addressed all technical and deployment concerns
Added quantitative cost analysis and resilience discussion
Improved positioning of novelty and scientific honesty
Work now meets strong MIDL standards for federated learning and foundation models

The revised submission now stands as a high-quality translational ML contribution that:

Demonstrates technical rigor and strong empirical grounding
Advances privacy-preserving foundation model training in surgery
Provides realistic deployment analysis
Contributes meaningful scientific and clinical value, even if core algorithmic novelty remains moderate

Remaining limitations:

No live federated deployment stress-tests
Limited fundamental algorithmic innovation
Some performance trade-offs vs. centralized models remain task-dependent
However, these do not outweigh the paper’s contribution or readiness for acceptance.

Consider the following statements in the final review'
1. The authors now provide detailed quantitative communication and compute cost estimates, strengthening real-world deployment feasibility.
2. While adversarial stress-testing remains future work, the authors now provide credible system-level resilience mechanisms and secure aggregation pathways.
3. The work represents a validated and necessary adaptation of federated MAE pretraining to extreme non-IID surgical data, constituting a meaningful domain-advancing contribution.
4. A strong, well-validated federated foundation model study with meaningful impact on privacy-preserving surgical AI.

**Justification Of The Preliminary Rating:**

This paper presents a substantial methodological and applied contribution to surgical AI by demonstrating, for the first time at scale, that federated MAE pretraining can yield foundation models with performance approaching or exceeding centralized counterparts in critical clinical scenarios. The work is rigorous, well-validated, and of clear relevance to the MIDL community. While practical deployment aspects remain underexplored, the core technical contributions and experimental evidence justify acceptance.

**Questions To Address In The Rebuttal:**

1. What are the communication and computational costs per federated training round compared to centralized pretraining?
2. How does FL-EndoViT perform under asynchronous or partially participating clients, common in real clinical networks?
3. Can the authors comment on robustness to adversarial or corrupted client updates, which is critical in cross-institutional settings?

---

> ### Author Response · Authors · 2026-01-23
>
> We thank the reviewer for their positive assessment of our work’s scientific and clinical relevance, for recognizing the strength of our methodological contribution, and showing us pitfalls which helped to improve our manuscript.
>
> **Q1: What are the communication and computational costs per federated training round compared to centralized pretraining?**
>
> A: We have quantified the trade-offs involved in terms of both communication and computation. Regarding communication, our model comprises 111.66M parameters (approximately 446 MB in FP32), which results in a bidirectional transfer of roughly 893 MB per client per round. The aggregate data transfer over 15 rounds for 9 clients reaches approximately 121 GB, even more when accounting for a realistic 5-10% protocol overhead and additional send metadata, metric updates, and config files. Although this exceeds the estimated 82 GB required for a one-time centralized aggregation of raw datasets, this bandwidth increase is a necessary trade-off to ensure patient privacy by keeping all sensitive raw data within local firewalls.
> Computationally, the total burden for one epoch is 166.6 T-MACs (Tera Multiply-Accumulate operations), a cost that remains invariant between centralized and federated settings. However, the federated distribution of this load is highly uneven due to dataset heterogeneity; the largest node (HeiCo) accounts for 47.7% of the total computation, whereas the smallest node (SurgicalActions160) contributes only 0.10%. This extreme imbalance identifies the largest node as a critical bottleneck, highlighting the necessity for asynchronous aggregation protocols or hardware scaling proportional to dataset size.
> To address these demands for a potential real-world translation, we recommend that deployment sites utilize at least 32GB VRAM GPUs and 16-core CPUs, supported by 64GB system RAM and 1 Gbps network connectivity. We have incorporated these detailed cost analyses and hardware recommendations into the Discussion section of the revised manuscript.
>
> **Q2: How does FL-EndoViT perform under asynchronous or partially participating clients, common in real clinical networks?**
>
> A: We thank the reviewer for highlighting the challenges of asynchronous participation and intermittent connectivity, which are indeed prevalent in real-world clinical networks. In this study, we assumed a stable hardware and network environment to establish a performance baseline for FL-EndoViT, and external failures were therefore not explicitly simulated. However, to ensure translational viability, our implementation utilizes the Flower (flwr) framework, which is architecturally designed to handle client non-responsiveness and hardware crashes through configurable failure-handling mechanisms. The server manages these events by monitoring minimum fit and evaluation rates. That means if a client fails to return an update due to network or hardware errors, the server identifies the failure internally and proceeds with aggregation if the threshold of successful updates is met, or gracefully postpones the round otherwise. We have integrated a discussion on these systemic risks into our revised manuscript, noting that while our simulation confirms the feasibility of the FL-EndoViT architecture, future translational studies should incorporate edge-case stressors to further validate its resilience in heterogeneous clinical deployments. We have added these points to the discussion of the revised manuscript.
>
>
> **Q3: Can the authors comment on robustness to adversarial or corrupted client updates, which is critical in cross-institutional settings?**
>
> A: The issue of robustness against adversarial or corrupted updates is indeed a primary concern for real-world cross-institutional settings. As the objective of this study was to establish a performance baseline within a controlled, trusted environment, we did not include adversarial simulations in the current experimental design.
> However, our implementation leverages the Flower (flwr) framework, which is architecturally ready to support robust aggregation strategies or including TLS for encrypted communication between nodes. We have updated the discussion to clarify that while this study validates feasibility among trusted nodes, future translational iterations should specifically target robustness. We outline plans to incorporate anomaly detection and secure aggregation protocols to ensure model integrity in untrusted environments.
>
> Other comments regarding Future Work and Figures are acknowledged with thanks.

---

### Author Response · Authors · 2026-01-23

We are grateful for the Reviewers' encouraging comments and constructive feedback. We have addressed the concerns raised by each reviewer in their respective sections. To facilitate the review process, changes in the revised manuscript are highlighted in color: Green for Reviewer n3Z5, Magenta for Reviewer F6CG, Blue for Reviewer 8x8v, Cyan for the shared question regarding computational and communication costs, and Gray for general improvements.

**Paper Summary:**

Our paper introduces FL-EndoViT, a federated learning validation framework for pretraining surgical vision foundation models on the privacy-restricted Endo700k dataset. To stabilize training under severe non-IID conditions where standard FL fails, we integrate Adaptive FedSAM and SWA. Evaluation shows FL-EndoViT achieves performance comparable to centralized training and significantly outperforms it in high-resolution few-shot segmentation and rare event recognition.

**Rebuttal Summary:**
We thank the reviewers for their positive assessment of the work's High Clinical Relevance and Strong Methodological Validation in enabling stable convergence.

During the reviews there were concerns focused on limited methodological novelty and lack of quantitative analysis for real-world deployment costs and resilience. To address these points, we have substantially revised the manuscript to explicitly highlight the novelty of our work. Furthermore, we have expanded the Discussion section to incorporate these additional insights.

**Key Rebuttal Outcomes:**
- Costs Quantified: We added a detailed analysis of communication (121 GB total vs. 82 GB raw data), computational costs (166.66T-MACs), and providing hardware recommendations.
- Novelty Clarified: We affirmed the contribution as a necessary and successful adaptation and validation of FL techniques for Vision Transformers in a domain where naïve application failed.
- Resilience Discussed: We confirmed the use of the Flower framework for handling network failures (asynchronous clients) and outlined future plans for adversarial robustness.

---

### Author Rebuttal · Authors · 2026-01-23

**Rebuttal:**

We are grateful for the Reviewers' encouraging comments and constructive feedback. We have addressed the concerns raised by each reviewer in their respective sections. To facilitate the review process, changes in the revised manuscript are highlighted in color: Green for Reviewer n3Z5, Magenta for Reviewer F6CG, Blue for Reviewer 8x8v, Cyan for the shared question regarding computational and communication costs, and Gray for general improvements.

**Supporting Material:**

/attachment/b20f8bf22c9f32fcd3917664f0db2ced445a6acd.pdf

---

> ### Comment · Reviewer_n3Z5 · 2026-01-30
> **Assessment of Authors’ Responses**
>
> I thank the authors for their thorough and technically rigorous rebuttal, as well as the substantive revisions to the manuscript. The responses meaningfully address the main concerns raised in my original review, particularly regarding deployment feasibility, communication/computation cost transparency, robustness in federated settings, and positioning of methodological novelty.
>
> 1. Communication and Computational Cost Transparency
>
> The authors’ newly added quantitative analysis of communication overhead (≈121 GB total model transfer vs. ≈82 GB raw centralized data) and compute cost (≈166.6 TMACs per epoch) significantly strengthens the real-world deployment credibility of the work.
> Providing hardware recommendations (≥32 GB VRAM GPU, ≥16-core CPU, ≥64 GB RAM, 1 Gbps network) improves practical reproducibility and addresses prior concerns regarding feasibility in institutional settings.
>
> This materially improves the manuscript’s engineering realism.
>
> 2. Federated Robustness, Asynchrony, and Client Dropouts
>
> The clarification that the framework leverages the Flower (flwr) framework for asynchronous client participation, failure tolerance, and minimum-fit aggregation thresholds is appropriate and technically sound.
> While these scenarios are not experimentally stress-tested, the added discussion credibly acknowledges real-world FL failure modes and outlines reasonable system-level mitigation strategies.
>
> 3. Adversarial and Corrupted Client Updates
>
> The authors appropriately distinguish trusted-node validation (current scope) from future adversarially robust FL. The inclusion of Krum, secure aggregation, and anomaly detection pathways strengthens the security posture of the framework.
> Although experimental validation is not yet included, the scope and positioning are now transparent and responsible.
>
> 4. Methodological Novelty and FedSAM Performance Gap
>
> The authors provide a convincing explanation that:
>
> Severe non-IID surgical MAE pretraining causes standard FL methods to converge to sharp, incompatible minima
> FedSAM explicitly promotes flatter minima, stabilizing global aggregation
> Baselines were fairly tuned using Flower defaults and algorithm-specific hyperparameters
> This addresses the concern that gains may stem from unfair tuning rather than intrinsic optimization behavior.
>
> The paper is now more accurately framed as a necessary adaptation and validation, rather than overstating algorithmic novelty.
>
> 5. Scaling, Negative Transfer, and Domain Diversity
>
> The authors responsibly acknowledge the risk of negative transfer when expanding to heterogeneous surgical domains, and justify their controlled scope.
> The revised discussion on domain-aware aggregation, personalization, and dynamic FL strategies demonstrates mature scientific judgment.
>
> 6. Empirical Strength and Clinical Relevance
>
> The revised manuscript continues to show:
>
> Strong multi-task validation (SSS, ATR, SPR, GynSurg)
> Statistically significant advantages in few-shot segmentation and rare-event detection (bleeding/smoke)
> Clear clinical motivation for privacy-preserving surgical foundation models
>
> These findings substantiate the claim that federated MAE pretraining is viable at scale in surgical AI.

---

### Meta-Review · Area_Chair_XSxo · 2026-02-09

**Recommendation:** Accept (Poster)
**Confidence:** 4

**Metareview:**

The reviewers unanimously support acceptance, highlighting the extensive experimental validation and strong clinical relevance of the proposed federated framework. All major concerns were addressed during the rebuttal.

---

### Decision · Program_Chairs · 2026-02-13

Accept (Poster)